# RESuM: Rare Event Surrogate Model for Physics Detector Design

**Ann-Kathrin Schuetz**[1]
aschuetz@lbl.gov

**Alan W. P. Poon**[1]
awpoon@lbl.gov

**Aobo Li**[2*]
aol002@ucsd.edu

[1]Nuclear Science Division, Lawrence Berkeley National Laboratory, Berkeley, CA 94720, USA
[2]Halıcıoğlu Data Science Institute, Department of Physics, UC San Diego, La Jolla, CA 92093, USA
* Corresponding Authors

## Abstract

The experimental discovery of neutrinoless double-beta decay (NLDBD) would answer one of the most important questions in physics: Why is there more matter than antimatter in our universe? To maximize the chances of detection, NLDBD experiments must optimize their detector designs to minimize the probability of background events contaminating the detector. Given that this probability is inherently low, design optimization either requires extremely costly simulations to generate sufficient background counts or contending with significant variance. In this work, we formalize this dilemma as a Rare Event Design (RED) problem: identifying optimal design parameters when the design metric to be minimized is inherently small. We then designed the Rare Event Surrogate Model (RESuM)[1] for physics detector design optimization under RED conditions. RESuM uses a pretrained Conditional Neural Process (CNP) model to incorporate additional prior knowledges into a Multi-Fidelity Gaussian Process (MFGP) model. We applied RESuM to optimize neutron moderator designs for the LEGEND NLDBD experiment, identifying an optimal design that reduces neutron background by $(66.5 \pm 3.5)\%$ while using only 3.3% of the computational resources compared to traditional methods. Given the prevalence of RED problems in other fields of physical sciences, the RESuM algorithm has broad potential for simulation-intensive applications.

## 1 Introduction

Why is there more matter than antimatter in our universe? This question remains one of the most important yet unsolved questions in physics. Several Nobel Prizes have been awarded for groundbreaking discoveries that have advanced our understanding of this questions, including the discovery of CP violation in kaons (Cronin and Fitch, 1980), the detection of cosmic neutrinos (Koshiba, 2002), and the development of the Kobayashi-Maskawa theory of CP violation (Kobayashi and Maskawa, 2008). Despite these monumental achievements, the reason for the dominance of matter over antimatter remains unsolved. One of the most promising next steps toward answering this question is the potential discovery of Neutrinoless Double-Beta Decay (NLDBD) (Dolinski et al., 2019). Such a discovery would represent a major milestone in this direction and would undoubtedly be considered a Nobel-Prize-level breakthrough in physics. Due to its utmost importance, the entire U.S. nuclear physics community has gathered for a year-long discussion in 2023 and recommended the experimental search for NLDBD as the second-highest priority (Committee, 2023) for next 10 years.

The most challenging aspect of NLDBD search is dealing with background events: physical events that are not NLDBD, but are indistinguishable from it. Since NLDBD is hypothesized to occur less than once every three years (LEGEND-Collaboration et al., 2021; Dolinski et al., 2019), even a single background event entering the detector could potentially ruin the entire detection effort. Therefore, designing ultra-pure NLDBD detectors with optimal parameters to minimize the probability of background events entering the detector becomes the utmost goal of all NLDBD experiments.

---

[1]Github Repository: https://github.com/annkasch/resum-legend

Traditionally, the detector design procedure is conducted with simulations: we first simulate our detectors and $N_1$ background events under a certain design parameter $\boldsymbol{\theta}_1$, then count the number of background events that eventually enter our detector, $m_1$. We then repeat the simulation process with another design parameter $\boldsymbol{\theta}_2$ and count $m_2$. If $m_1/N_1 < m_2/N_2$, it suggests that the design $\boldsymbol{\theta}_1$ is better than $\boldsymbol{\theta}_2$. This simulation process can be repeated multiple times until an optimal design is found. An obvious shortcoming of this traditional approach is the computational cost: due to the ultra-pure nature of the NLDBD detector, $N$ needs to be very large for $m$ to even be non-zero. This is amplified by the complexity of the design space, involving numerous and often non-linearly inter-dependent parameters such as detector geometry, material properties, and environmental conditions.

An obvious solution to this problem is to build a surrogate model that can significantly accelerate our simulations (Li et al., 2023; Ravi et al., 2024). However, due to the rare event nature, $m$ is either 0 or a small, discrete integer, which leads to high variance in our design metric $m/N$. This variance renders training traditional continuous surrogate models extremely difficult. In this paper, we formulate this problem as a Rare Event Design (RED) problem and present RESuM—a Rare Event Surrogate model to solve this problem. RESuM navigate through a complex landscape and approximate the complex relationships between the design parameters $\boldsymbol{\theta}$ and rare event design metric $m/N$. The benchmarking result shows that RESuM could reduce the LEGEND neutron background by $(66.5\pm3.5)\%$ using only 3.3% of the computational power compared to traditional methods. Due to the broad presence of RED problems in physical sciences, RESuM has the potential to be applied to other domains, including Astronomy and Material Science.

## 2 RELATED WORKS

Due to the computational cost of particle physics simulations, generative models like VAE (Z. Fu et al., 2024), GAN (Kansal et al., 2021; Vallecorsa, 2018; Hashemi et al., 2024), and diffusion models are widely used as surrogate models for fast simulation (Kansal et al., 2023). Although these deep generative models, usually trained on large datasets, robustly reproduce enriched high-dimensional data, their black-box nature renders them non-interpretable and lacking clear statistical meaning. Meanwhile, the CNP model (Garnelo et al., 2018), as a probabilistic generative model, offers the distinct advantage of few-shot learning and provides clear statistical interpretation. It has demonstrated good performance in few-shot problems, including classification tasks (Requeima et al., 2019), statistical downscaling (Vaughan et al., 2022), and hydrogeological modeling (Cui et al., 2022). In this study, we explore a novel surrogate modeling approach that focuses solely on key detector design metrics. The CNP model was not used as a generative model, but as a predictive model to smooth out discreteness of rare design metrics.

Another related field is rare event simulation and modeling in reliability engineering. The rare event problem here focuses on estimating extremely low failure probabilities $P_f$. Since direct Monte Carlo simulation becomes intractable as $P_f$ approaches zero, specialized techniques, including adaptive sampling and FORM/SORM methods, have been developed. The development progressed from FORM by Hasofer (1974) and its extension to non-normal distributions (Fiessler et al., 1979), comprehensively reviewed in Der Kiureghian et al. (2005). Methodological advances include adaptive sampling (Bucher, 1988), surrogate-based methods (Li and Xiu, 2010; Li et al., 2011), sequential importance sampling (Papaioannou et al., 2016), and multi-fidelity approaches (Peherstorfer et al., 2016; 2018). Recent work has introduced multilevel sampling (Wagner et al., 2020) and ensemble Kalman filters (Wagner et al., 2022). While Adaptive Importance Sampling (AIS) can potentially solve the RED problem, its implementation to LEGEND simulation presents several significant challenges, as discussed in Appendix 15. In contrast, the RESuM model proposed in this work provides a simple yet efficient solution to the RED problem in LEGEND.

## 3 RARE EVENT DESIGN PROBLEM

**Definition** Let $\boldsymbol{\theta} \in \Theta$ be the vector of design parameters, where $\Theta$ represents the space of all possible design parameters. Consider a simulation involving $N$ events, or data points, under design parameter $\boldsymbol{\theta}$; each event can either trigger a signal [2] or not. Define a stochastic process $\{X_1, \ldots, X_N\}$,

---

[2]"trigger a signal" could represent any event of interest depending on the task setup. In the case of the NLDBD background minimization task, it means a background event successfully leaches into the detector

where each random variable $X_i$ corresponds to the $i^{th}$ event in the simulation and $X_i = 1$ if the $i^{th}$ event triggers a signal and $X_i = 0$ if it doesn't. Each random variable $X_i$ is statistically independent of all other $X_j$ for $j \neq i$.

Each simulated event $i$ is considered independent, and the outcome of $X_i$ depends on two sets of parameters: a set of design parameters $\boldsymbol{\theta}$ which is universal across all events, and another sets of event-specific parameters $\boldsymbol{\phi_i} \in \boldsymbol{\Phi}$ where $\boldsymbol{\Phi}$ represents the space of all possible event-specific parameters. The probability that the $i$-th event will trigger a signal is thereby defined as a function of both $\boldsymbol{\theta}$ and $\boldsymbol{\phi_i}$, which could be denoted as $t(\boldsymbol{\theta}, \boldsymbol{\phi_i})$.

Let $m$ represent the number of events that trigger a signal. The design metric $y$ is then defined as:

$$y = \frac{m}{N} = \frac{\sum_{i=1}^{N} X_i}{N} \tag{1}$$

**Rare Event Assumption** The number of triggered events $m$ follows a binomial distribution with the triggering probability $t(\boldsymbol{\theta}, \boldsymbol{\phi_i})$. Under the rare event assumption that $m \ll N$ and the triggering probability for each event $t(\boldsymbol{\theta}, \boldsymbol{\phi_i})$ is small, the number of triggered events $m$ can be approximated by a Poisson distribution as $m \sim \text{Poisson}\left(N\bar{t}(\boldsymbol{\theta})\right)$. Where $\bar{t}(\boldsymbol{\theta})$ is the expected triggering probability over all simulated events when $N$ goes to infinity:

$$\bar{t}(\boldsymbol{\theta}) = \int t(\boldsymbol{\theta}, \boldsymbol{\phi})g(\boldsymbol{\phi})d\boldsymbol{\phi} \tag{2}$$

The function $g(\boldsymbol{\phi})$ denotes a predefined probability density function (PDF) where $\phi_i$ could be sampled from during the simulation process. $\bar{t}(\boldsymbol{\theta})$ is obtained by marginalizing $t(\boldsymbol{\theta}, \boldsymbol{\phi})$ over $g(\boldsymbol{\phi})$. Therefore, the ultimate metric that we want to minimize is $\bar{t}$, which is the expectation of $y$:

$$\boldsymbol{\theta}^* = \arg\min_{\boldsymbol{\theta} \in \boldsymbol{\Theta}} \bar{t}(\boldsymbol{\theta}) \tag{3}$$

Since $\bar{t}$ depends on $\boldsymbol{\theta}$, minimizing $\bar{t}$ requires extensive sampling of different $\boldsymbol{\theta}$ values within the design space $\boldsymbol{\Theta}$ to identify the optimal parameter.

**Large N Scenario** Assuming that $\bar{t}(\boldsymbol{\theta})$ remains fixed. When $N$ becomes large, according to the central limit theorem, $m$ will tend to follow a normal distribution:

$$m \sim \mathcal{N}(N\bar{t}(\boldsymbol{\theta}), N\bar{t}(\boldsymbol{\theta}))$$

Since $y = m/N$, this means that $y$ will also follow a normal distribution with symmetric, well-defined statistical uncertainties $\bar{t}(\boldsymbol{\theta})/N$:

$$y \sim \mathcal{N}(\bar{t}(\boldsymbol{\theta}), \bar{t}(\boldsymbol{\theta})/N)$$

As $N \to +\infty$, $y$ will asymptotically approximate $\bar{t}(\boldsymbol{\theta})$ with statistical uncertainties approaching 0.

**Small N Scenario** When $N$ becomes small, the total number of instances $m$ that trigger a signal has a higher variance, as each individual instance has a significant impact on $m$. The accuracy measure $y = \frac{m}{N}$ can no longer be approximated with a normal distribution. This makes $y$ more sensitive to statistical fluctuations of a few simulated events. Furthermore, there is a non-negligible probability that no event will trigger a signal, meaning that $m = 0$ and $y \sim \frac{m}{N} = 0$. In summary, in the small N scenario, the design metric $y$ of interests will only takes on a discrete set of values, $y \in \left\{\frac{0}{N}, \frac{1}{N}, \ldots, \frac{m}{N}\right\}$.

## 4 RARE EVENT SURROGATE MODEL

The Rare Event Surrogate Model (RESuM) aims to solve the RED problem under the constraint of limited access to large $N$ simulations and an unknown triggering probability $t(\boldsymbol{\theta}, \boldsymbol{\phi_i})$. Consider a scenario where we run $K$ simulation trials with different design parameter $\boldsymbol{\theta}$, indexed by $k$; each simulation trial contains $N$ events indexed by $i$. The RESuM model includes three components: a Conditional Neural Process (CNP) (Garnelo et al., 2018) model that is trained on event level; a Multi-Fidelity Gaussian Process (MFGP) (Kennedy and O'Hagan, 2000; Qian and Wu, 2008) model that trains on simulation trial level; and active learning techniques to sequentially sample the parameter space after training. The conceptual framework and details of our model design are outlined in the following subsections.

### 4.1 BAYESIAN PRIOR KNOWLEDGE WITH CONDITIONAL NEURAL PROCESS

The random variable $X_{ki}$ represents whether the $i^{\text{th}}$ event triggered a signal or not. In traditional particle physics, the value of $X_{ki}$ is determined through a Monte Carlo simulation process: first, a parameter $\phi_{ki}$ is sampled from the distribution $g(\phi)$ to generate the event. This event then propagates through the detector, characterized by the design parameter $\theta_k$. The outcome of the simulation, which implicitly involves the joint distribution $t(\theta_k, \phi_{ki})$, is only observed as $X_{ki}$. As discussed before, $X_{ki}$ can only be 0 or 1. In the small $N$ scenario, the root cause of the discreteness of $y$ is this binary nature: 1 if a signal is triggered or 0 if not. This produces significant statistical variance in $y$. Suppose we want to model this simulation process with a Bernoulli distribution:

$$X_{ki} \sim \text{Bernoulli}(t(\theta_k, \phi_{ki})) \tag{4}$$

The goal of incorporating prior knowledge is to smooth out the binary $X_{ki}$ into a continuous, floating-point score $\beta$ between 0 and 1. The score $\beta_{ki}$ should approximate $t(\theta_k, \phi_{ki})$ given design parameter $\theta_k$ and event-specific parameter $\phi_{ki}$.

This work provides an alternative solution by adopting a similar idea to the CNP model. CNP works by learning a representation of input-output relationships from context data to predict outputs for new inputs (Garnelo et al., 2018). In our case, the input is $\theta_k$ and $\phi_{ki}$, and the output is the random variable $X_{ki}$. The random process that generates $X_{ki}$ based on the inputs is the Bernoulli process controlled by $t(\theta, \phi)$. We then adopt the same representation learning idea used in the CNP, which involves approximating the random process by sampling from a Gaussian distribution conditioned at observed data. The mean and variance are modeled with neural networks:

$$\text{Bernoulli}(t(\theta, \phi)) \approx \text{Bernoulli}(\beta) \tag{5}$$

$$\beta \sim \mathcal{N}(\mu_{NN}(\theta, \phi, w), \sigma^2_{NN}(\theta, \phi, w))|_{X_{ki}, \phi_{ki}, \theta_k} \tag{6}$$

Where $\beta$ is the CNP-generated score in general and $\beta_{ki} = \beta|_{\theta_k, \phi_{ki}}$ is the CNP score of a specific event ($i^{th}$ event in the $k^{th}$ simulation trial). The nuisance parameters $w$ represent the trainable parameters of the neural network (Garnelo et al., 2018), including the weights and biases, that are optimized during training by minimizing the likelihood of the observed data. Importantly, the neural networks are not trained to predict the binary observable $X$, but rather to estimate the continuous floating-point score $\beta$. A comprehensive description of the CNP model, along with the interpretation of the score $\beta$ and the associated loss function (likelihood), is provided in Appendix 13. The score $\beta$ for each simulated event serves as prior information that is incorporated into the MFGP surrogate model.

### 4.2 MODEL DESCRIPTION

Building on the conceptual framework described in 4.1, we will provide an end-to-end overview of RESuM as shown in Figure 1. We generate two types of simulations: low-fidelity (LF) and high-fidelity (HF). Detailed descriptions of these simulations can be found in Section 5.1. The primary distinction between them lies in the number of simulated events $N$, where $N_{HF} \gg N_{LF}$. Another key difference is the distribution $g(\phi)$ from which the parameter $\phi_i$ of each event is sampled, where HF simulation contains a more complicated, physics-oriented $g(\phi)$. The low computational cost of LF simulation allows us to simulate more trials thereby exploring a broader range of $\theta$. The first step is to train the CNP model. The CNP comprises three primary components: an encoder, an aggregator, and a decoder. The parameters $\theta_k, \phi_{ki}$, and $X_{ki}$ of each simulated event are first concatenated into a context point. The encoder, implemented as a Multi-Layer Perceptron (MLP), transforms each context point into a low-dimensional representation. These representations are then aggregated through averaging to form a unified representation that represents $t(\theta)$. The decoder uses $t(\theta)$ and the $\phi_{ki}$ of new data to output parameters $\mu_{ki}$ and $\sigma^2_{ki}$ for each event $i$. We then use $\mu_{ki}$ and $\sigma^2_{ki}$ to form a normal distribution and sample a CNP score $\beta_{ki}$ from it. The scores $\beta_{ki}$ are chosen to naturally fit a normal-like distribution but bounded between 0 and 1. Since the CNP is trained at event level, $\beta_{ki}$ will be the same regardless of whether the event is generated in HF or LF simulation.

Based on the trained CNP model, the next step involves in calculating three design metrics at different fidelities. The first one is $y_{Raw} = m/N$ from HF simulations, which is the ultimate design

Figure 1: Overview of the RESuM framework for solving RED problems. The left side illustrates the CNP used for modeling both LF and HF simulation data. The CNP aggregates event-specific parameters $\phi_i$ and design parameters $\theta$ from LF and HF simulations to produce $y_{\text{CNP}}^{\text{LF}}$ and $y_{\text{CNP}}^{\text{HF}}$, which, together with HF simulation output $y_{\text{Raw}}^{\text{HF}}$, serve as inputs to the surrogate model. The right side shows the multi-MFGP that combines predictions $\hat{y}_{\text{CNP}}$ from LF and HF to estimate the HF design metric $\hat{y}_{\text{Raw}}^{\text{HF}}$.

metric we want our surrogate model to emulate. The second metric is also derived from HF simulations but is defined as the average CNP score of all simulated events:

$$y_{CNP} = \frac{1}{N} \sum_{i=0}^{N} \beta_{ki} \tag{7}$$

The third metric is $y_{CNP}$ calculated over LF simulations. These three design metrics are then incorporated into a MFGP model to train the surrogate model. Co-kriging was used to account for correlations among different design metrics. The mathematical detail of MFGP can be found in Appendix 11.

After training the MFGP model, we adopt active learning to select new sampling points $\theta_{\text{new}}$ to generate $y_{Raw}$ with HF simulations. Since HF simulation is expensive, to determine which point to collect next, we use a gradient-based optimizer to find $\theta_{n+1} = \arg\max_{\theta \in \mathbb{X}} \mathcal{I}(\theta)$ (Paleyes et al., 2023). The acquisition function $\mathcal{I}(\theta)$ determines the next data point to explore by balancing exploration (high variance) and exploitation (high mean). We chose the integrated variance reduction method, where the next point, $\theta_{n+1}$, is selected to maximally reduce the total variance of the model (Sacks et al., 1989). More detail about the active learning method can be found in Appendix 12.

## 5 EXPERIMENT AND RESULT

The Large Enriched Germanium Experiment for Neutrinoless Double-Beta Decay (LEGEND) is a next-generation pioneering experiment in the search for NLDBD, with over 300 international collaborators. One of the major background event type in LEGEND are $^{77(\text{m})}\text{Ge}$, which are produced through a three-step physics process: (1) **Cosmic Muons** are high-energy particles that constantly shower down from the sky. (2) When cosmic muons enter the LEGEND outer detector, they can interact with materials in the outer detector, which generates a lot of **neutrons**. (3) Neutrons then propagate through the LEGEND detector system. If a neutron enters the inner detector, it has a chance to produce $^{77(m)}\text{Ge}$ by neutron capture, which is the primary background of concern. $^{77(\text{m})}\text{Ge}$, once produced, will be particularly challenging, because it could mimic NLDBD events, making it nearly impossible to distinguish and reject once produced[3]. The most viable solution to mitigate this background is through the design of a neutron moderator—a neutron shield that slows down the neutrons and reduce the neutron flux entering the inner detector system between step (2) and (3). Figure 2 provides an overview of the LEGEND detector and a proposed neutron moderator design. Our goal with the RESuM is to optimize the geometric design of the neutron moderator to prevent most neutrons from leaching into the inner detector. The optimization process consists of two key steps: generating simulations under different design parameters, and adopting the RESuM model to surrogate and identify the optimal design of neutron moderators.

---

[3]Currently, there are no efficient methods to eliminate $^{77(\text{m})}\text{Ge}$ once created, aside from employing complex active tagging algorithms (Neuberger et al., 2021) that introduce additional dead time.

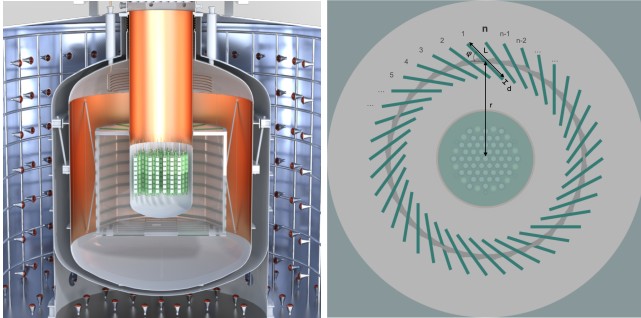

Figure 2: Left: Rendering of the LEGEND-1000 experiment with the hundreds of detectors. The inner detector region is shown in green while the rest are outer detectors. Right: Illustration of the neutron moderator/shield design with the 5 design parameters (viewed from top).

## 5.1 NEUTRON MODERATOR SIMULATIONS

Two neutron moderator design schemes were proposed: (1) a continuous, cylindrical-shaped shield surrounding the detector array, and (2) a turbine-like shield composed of several panels. As illustrated in Figure 2 (Right), we define five design parameters to include both geometries under continuous transition: the radius $r$ of the cylindrical layer or the distance of the panels, the shield thickness $d$, the number of panels $n$, the angle of the panels $\varphi$, and the panel length $L$. These five parameters constitute the design parameter space $\Theta$, where each parameter is allowed to vary within a predefined range.

A specific design $\theta$ can be sampled from $\Theta$ to perform simulations and obtain the corresponding design metric $y_{Raw}$. For a given $\theta$, we utilized a Monte Carlo (MC) simulation package based on the GEANT-4 toolkit (Agostinelli et al., 2003; Allison et al., 2006), integrated with the existing LEGEND software frameworks (Neuberger; Ramachers and Morgan). We first simulate the LEGEND detector with a neutron moderator configuration according to the sampled design parameter $\theta$, as shown in shown in Figure 2. Second, we simulate neutrons which are allowed to propagate through the detector within the simulation system, and the number of $^{77(m)}$Ge produced, $m$, is counted within the inner detector. The design metric $y_{Raw}$ is then computed as $y_{Raw} = m/N$.

We implemented two levels of simulation fidelity: high-fidelity (HF) and low-fidelity (LF). Since neutrons are primarily produced by cosmic muons showering down from the atmosphere, the HF simulation starts by generating muons outside the LEGEND detector using physics information provided by the MUSUN muon simulation software (Kudryavtsev, 2009). The event-specific parameter $\phi_\mu$ of each muon is sampled from a carefully-designed distribution $g_\mu(\phi_\mu)$. These muons, with predefined energies and directions, will propagate through the detector geometry, leading to the production of neutrons with their associated $\phi_n^{ini}$ assigning an initial position $(x, y, z)$, initial momentum $(p_x, p_y, p_z)$, and initial energy $E$ to the created neutrons. The total number of simulated muons $N_{HF}$ for each design parameter $\theta_k$ is typically very large, on the order of $10^7$. The LF simulation, on the other hand, skipped the muon simulation step. Neutrons are directly simulated within the detector, and the event-specific parameter $\phi_n^{ini}$ is randomly sampled from a uniform distribution $g_n^{ini}(\phi_n^{ini})$ in a predefined range. The total number of neutrons simulated, $N_{LF}$ is on the order of $10^4$, which is also significantly smaller than in the HF simulation. To fully assess the probability of a neutron in step (3) producing a $^{77(m)}$Ge background event, we must also consider the neutron's end position, the time from its creation to termination, and its energy transfer at termination. Consequently, ten parameters form the event-specific parameter $\phi = (\phi_n^{ini}, \phi_n^{fin})$ obtained from a predefined distribution $g(\phi) = \int g_n^{fin}(\phi_n^{fin} \mid \phi_n^{ini}) \cdot g_n^{ini}(\phi_n^{ini} \mid \phi_\mu) \cdot g_\mu(\phi_\mu) \, d\phi_\mu$ (as discussed in Section 3).

While the LF simulation is based on a simplified distribution $g_n^{ini}(\phi_n^{ini})$ (instead of $g_n^{ini}(\phi_n^{ini} \mid \phi_\mu)$), it offers significant computational advantages, with a cost of only 0.15 CPU hours per run—about 1130 times faster than the HF simulation. This allows a broader exploration in the design parameter space $\Theta$. Conversely, the HF simulations are crucial for providing rigorous estimates of the $^{77(m)}$Ge background event production rate, ensuring that the optimized designs meet the stringent

background requirements for the LEGEND experiment. In total, 4 HF and 304 LF simulations were generated to form the training dataset for the surrogate model, with the four HF samples being reused from a previous simulation study, as suggested in Neuberger et al. (2021).

## 5.2 CONDITIONAL NEURAL PROCESS RESULT

The network structure and motivation of the CNP are discussed in Section 4.2. Training is performed using supervised learning, where a signal label (1) is assigned to neutrons that successfully produce $^{77(\mathrm{m})}$Ge background, and a background label (0) is assigned to neutrons that do not. The network input parameter $x$ here consists of the design parameters $\boldsymbol{\theta}_k$, as well as the event-specific parameters $\phi_{\mathrm{n},ki}^{\mathrm{ini}}$ and $\phi_{\mathrm{n},ki}^{\mathrm{fin}}$. A major challenge in training the CNP is the severe imbalance between signal and background, with a ratio of approximately $1 : 5 \cdot 10^4$. This is consistent with the rare event assumption, where $m \ll N$. To address this imbalance, we apply mixup (Zhang et al., 2018) techniques which generates new training samples $\hat{x}$ by forming linear combinations of existing signal samples $x_l$ and background samples $x_j$, along with their corresponding labels $y_l$ and $y_j$:

$$\hat{x} = \lambda x_l + (1 - \lambda)x_j \quad \text{and} \quad \hat{y} = \lambda y_l + (1 - \lambda)y_j$$

where $\lambda$ is randomly drawn from a beta distribution $B(0.1, 0.1)$. This process introduces a weighted blend of signal and background, helping to alleviate the imbalance in the data and improve the model's generalization and robustness. To demonstrate the effectiveness of the CNP in reducing

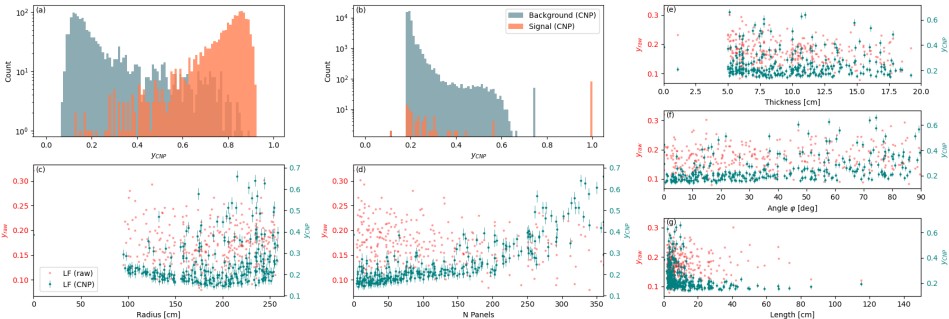

Figure 3: Comparison of raw data (red points) and CNP predictions (green points with error bars) across different design parameters. a) CNP result on the training dataset with mixup. b) predicted scores for a validation dataset. c) to g) provides scatter plots of the raw metric $y_{\mathrm{Raw}}$ (red points) and $y_{\mathrm{CNP}}$ (green points with error bars), plotted against each design parameter in $\boldsymbol{\Theta}$. The CNP model offers smoother, more refined predictions compared to $y_{\mathrm{Raw}}$.

statistical variance in the design metric, we calculated two different metrics for each LF simulation trial: the raw design metric $y_{\mathrm{Raw}}$ (in $[\mathrm{nuclei}/(\mathrm{kg} \cdot \mathrm{yr})]$) and the averaged CNP score $y_{\mathrm{CNP}}$. The results of both metrics are plotted against the five design parameters in Figure 3. As anticipated, the raw metric $y_{\mathrm{Raw}}$ exhibits significant statistical fluctuations that overshadows any correlations with respect to each design parameter. In contrast, the CNP score $y_{\mathrm{CNP}}$ reveals clear dependencies on the radius and number of panels (see Figure 3 Bottom). This indicates that CNP effectively reveal additional prior information into the MFGP.

## 5.3 RESuM RESULT

With CNP-generated scores, the full RESuM model was trained with three design metrics at different fidelities: $y_{\mathrm{Raw}}$ for HF simulations, $y_{\mathrm{CNP}}^{\mathrm{HF}}$ for HF simulations, and $y_{\mathrm{CNP}}^{\mathrm{LF}}$ for LF simulations. The ultimate goal is to emulate $y_{\mathrm{Raw}}$ for HF simulations which provides a more accurate representation of $^{77(\mathrm{m})}$Ge production rates under design parameter $\boldsymbol{\theta}$. The MFGP model was carried out by using the Emukit python library (Paleyes et al., 2023). Figure 4 (Left Bottom) illustrates the active learning process, including the HF model prediction (Top) and the acquisition function (Bottom). The surrogate model provides an estimate of the design metric of interest ($y_{Raw}$ from HF simulations) along with its associated uncertainty at each point in the input space. The acquisition function evolves as new data points refine the objective function approximation, balancing exploitation of known optima with exploration of uncertain regions. As notable in the lower panel of Figure 4 (Left), the

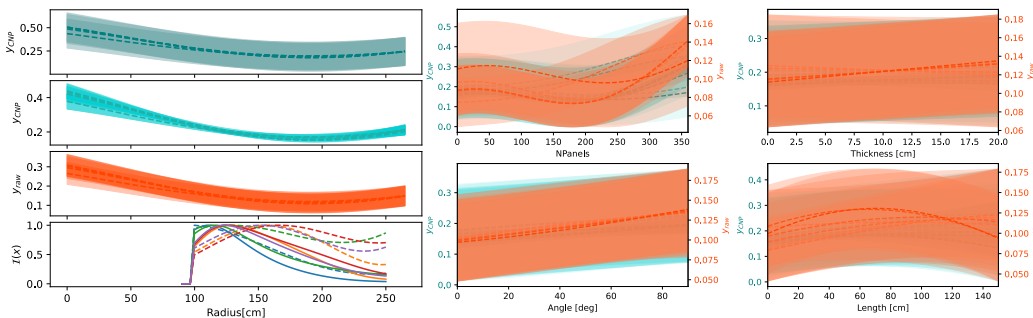

Figure 4: Left: One-dimensional CNP-LF (cyan), CNP-HF (dark cyan) and HF (orange) model predictions (dashed line) with uncertainty band (shaded area) as a function of the radius $r$. The lower panel shows the acquisition function as a function of the radius after each iteration. It guides the selection of future evaluation points in the input space to efficiently search for the optimal solution. Right: One dimensional model predictions over various design parameters. It illustrates the sequential model prediction update by adding new sampling points in each iteration.

acquisition function initially explores regions with high uncertainty, especially at medium distances where the optimum is likely to be found. The active learning procedure can be iterated as long as computation resources allow. In this work, six active learning iterations were performed to obtain the final result. The trends shown in Figure 4 reflect well-known detector physics: $r$ represents the moderator-detector distance, which determines the moderator's coverage volume, while $n$ denotes the number of moderator panels, where inter-panel gaps allow neutrons to escape. The dependence on other parameters are less obvious due to the schocastic nature of the system.

The HF model predictions for the $^{77(\mathrm{m})}$Ge production rate are shown in Figure 4, which displays one-dimensional projections of $y_{\mathrm{CNP}}^{\mathrm{LF}}$ from LF simulations (cyan), $y_{\mathrm{CNP}}^{\mathrm{HF}}$ from HF simulations (dark cyan), and $y_{\mathrm{Raw}}$ from HF simulations (orange) model predictions (dashed line) along with associated uncertainties (shaded area), as functions of 5 design parameters. These figures illustrate how the model's uncertainty decreased with each new sampling iteration. After six active learning iterations, the model predictions converged on several optimal designs, as shown in Table 1. These designs exhibit a range of configurations, with most favoring smaller panel angles $\varphi$ and a higher number of shorter panels, while one achieve optimal performance with fewer but significantly longer panels. Additionally, the optimal designs tend to cluster around two radii ranges, approximately 165 cm and 200 cm, suggesting that positioning the neutron moderator near these distances provides the best balance between effective neutron capture and material efficiency. This positioning allows for sufficient moderator mass to trap neutrons while maintaining an appropriate distance to minimize neutron escape. These designs outperformed those with larger gaps between panels, primarily due to the increased thickness of neutron moderator shielding. The optimal design reduces the $^{77(\mathrm{m})}$Ge production rate from $0.238 \ \mathrm{nuclei}/(\mathrm{kg} \cdot \mathrm{yr})$ to $0.0798 \ \mathrm{nuclei}/(\mathrm{kg} \cdot \mathrm{yr})$, leading to a $(66.5 \pm 3.5)\%$ reduction in neutron-induced background in LEGEND. Notably, RESuM identified the optimal de-

Table 1: Optimal neutron moderator design parameters identified by the RESuM model.

| Radius $r$ [cm] | Thickness $d$ [cm] | NPanels $n$ | Angle $\varphi$ [deg] | Length $L$ [cm] | $\hat{y}_{\mathrm{Raw}}^{min}$ [nuclei/(kg·yr)] | $\sigma_{\mathrm{Raw}}^{min}$ [nuclei/(kg·yr)] |
|---|---|---|---|---|---|---|
| 165.6 | 3.3 | 188 | 19.3 | 7.5 | 0.0798 | 0.0483 |
| 207.3 | 2.8 | 120 | 3.5 | 3.2 | 0.0786 | 0.0494 |
| 202.2 | 2.4 | 153 | 9.1 | 3.0 | 0.0787 | 0.0489 |
| 164.3 | 4.2 | 192 | 15.4 | 3.1 | 0.0784 | 0.0485 |

sign parameters with drastically reduced computational costs. Each HF simulation required 170 CPU hours, while each LF simulation needed just 0.15 CPU hour. If we were to explore the design space with only HF simulations, traditional methods would have required 52,700 CPU hours to explore all 310 design parameter sets. In contrast, RESuM used 310 LF simulations and 10 HF (4 for

MFGP training and 6 for active learning) simulations, totaling 1746.5 CPU hours—only 3.3% of the computational resources required by conventional approaches.

# 6 MODEL VALIDATION AND BENCHMARKING

The ultimate goal of this manuscript is to learn the functional mapping $f : \boldsymbol{\theta} \to y$. Each trained model should be able to take $\boldsymbol{\theta}$ as input and predict $\hat{y}_{\text{Raw}}$ as the predicted mean and $\hat{\sigma}_{\text{Raw}}$ as the prediction uncertainty. For this purpose, additional independent, out-of-sample HF simulations are generated as validation dataset, and the ground truth $y_{\text{Raw}}$ for each HF simulation is calculated by counting the number of background events $m$. Given the high computational demands of HF simulations and our limited resources, we generated 100 out-of-sample HF simulations at randomly sampled $\boldsymbol{\theta}$ values. We benchmarked the RESuM (MFGP+CNP) model against two baseline models: a Multi-Fidelity Bayesian Neural Network (MF-BNN,Bingham et al. (2019); Guo et al. (2022)), and a standard MFGP model Paleyes et al. (2023). Due to convergence speed difference, each model was trained on different dataset variants and validated on the same validation dataset. The comparison utilized five benchmarking metrics: the **Mean Square Error (MSE)** between $y_{\text{Raw}}$ and $\hat{y}_{\text{Raw}}$, statistical coverage at **1$\sigma$, 2$\sigma$, and 3$\sigma$** coverages, and computational **cost** to generate training data expressed as a percentage relative to traditional methods. Model selections and additional details are discussed in Appendix 14. The benchmarking result is shown in Table 2. For each trial, the

Table 2: Benchmarking Result of RESuM model with respect to 3 different baseline models.

| Trial | Model | Dataset (#LF,#HF) | MSE | $1\hat{\sigma}$ [%] | $2\hat{\sigma}$ [%] | $3\hat{\sigma}$ [%] | Cost [%] |
|---|---|---|---|---|---|---|---|
| 1 | MF-BNN | (305,5) | 0.471 | 100 | 100 | 100 | 1.7 |
| 2 | MF-BNN | (307,7) | 0.439 | 100 | 100 | 100 | 2.3 |
| 3 | MF-BNN | (310,10) | 0.021 | 100 | 100 | 100 | 3.3 |
| 4 | MFGP | (305,5) | 0.025 | 42 | 100 | 100 | 1.7 |
| 5 | MFGP | (307,7) | 0.007 | 31 | 56 | 79 | 2.3 |
| 6 | MFGP | (310,10) | 0.015 | 17 | 33 | 48 | 3.3 |
| 7 | AIS-MFGP | (305,5) | 0.002 | 39 | 66 | 80 | 1.7 |
| 8 | AIS-MFGP | (307,7) | 0.023 | 4 | 9 | 33 | 2.3 |
| 9 | AIS-MFGP | (310,10) | 0.001 | 39 | 64 | 86 | 3.3 |
| 10 | RESuM | (305,5) | 0.002 | 67 | 96 | 99 | 1.7 |
| 11 | RESuM | (307,7) | 0.003 | 64 | 92 | 99 | 2.3 |
| 12 | RESuM | (310,10) | 0.002 | 69 | 95 | 100 | 3.3 |
| 13 | RESuM (100x) | (310,10) | 0.003 | 62.4 | 92.2 | 99.6 | 3.3 |
| | Proper Statistical Coverage | | | 68.27 | 95.45 | 99.73 | |

coverage is computed by counting the percentage of ground truth $y_{Raw}$ that falls within the 1/2/3 $\hat{\sigma}_{Raw}$ band of $\hat{y}_{Raw}$. Assuming a standard normal distribution, the expected 1, 2, and 3$\hat{\sigma}$ coverages are 68.27%, 95.45%, and 99.73% , respectively. Among all benchmarked models, only the RESuM model (Trial 10-13) achieves both high prediction accuracy (low MSE) and proper statistical coverage. In comparison, the MFGP (Trial 4-6) and AIS-MFGP (Trial 7-9) models exhibit worse prediction accuracy and significant undercoverage. The MF-BNN model (Trial 1-3) yields much worse prediction accuracy with significant overcoverage. We also trained the RESuM model 100 times (Trial 13) with different randomly sampled datasets and validation sets. It shows that RESuM's prediction and coverage are consistently good. The coverage results, illustrated in Figure 5, underscore RESuM's capability to effectively surrogate complex detector design simulations.

# 7 LIMITATIONS AND APPLICATIONS

**Limitations:** The primary limitation of this work is the restricted computational resources available for thoroughly validating the model's performance. Due to these constraints, we were limited to generating only 100 HF simulations to assess coverage. Ideally, with unlimited computational resources, a comprehensive grid search across the 5-dimensional design parameter space would allow for more robust validation. Additionally, the active learning strategy employed in RESuM is relatively simplistic. Future work will focus on exploring more sophisticated active learning algorithms to more efficiently identify the optimal design.

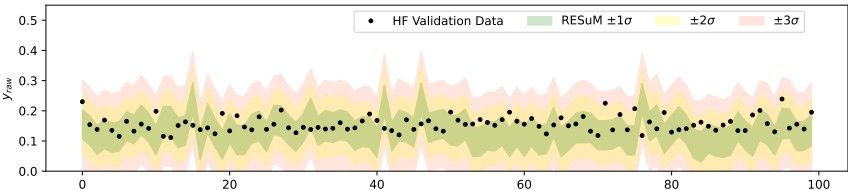

Figure 5: Coverage plot of the RESuM model predictions (Trial 9 in Table 5). The shaded regions represent the uncertainty bands at different confidence levels: $\pm 1\hat{\sigma}$ (green), $\pm 2\hat{\sigma}$ (yellow), and $\pm 3\hat{\sigma}$ (red). The RESuM model captures the overall trend of the HF validation data with proper coverage.

**Generalizability:** Although this work focuses on a specific detector design scenario within the LEGEND experiment, we believe that the mathematical formulation of RED problem, as outlined in Section 3, is applicable to a wide range of simulation and optimization challenges in the physical sciences. A few examples are provided below:

- **Astromony:** In computational astronomy, an emerging area involves simulating binary black hole (BBH) mergers to match the gravitational wave (GW) signals detected by the LIGO experiments (Fishbach and Holz, 2017). This process involves highly complex and computationally expensive many-body simulations (Kruckow et al., 2018). In this context, cosmological constants, such as the Hubble Constant and Dark Energy Density, can be treated as design parameters $\theta$, while the initial position, mass, and spin of each black hole are considered event-specific parameters $\phi$. The design metric $m$ is defined as the number of BBH mergers occuring over a given time period, with N representing the black holes simulated. The ground truth is provided by the GW catalog from LIGO. Some related work in this direction include (Lin et al., 2021; Broekgaarden et al., 2019).

- **Material Science**: First-principles simulations using Density Functional Theory (DFT) are fundamental in condensed matter research for material property prediction and design optimization (Dovesi et al., 2018; Hutter et al., 2014; Kang et al., 2019). While DFT simulations involve multiple parameters (temperature, pressure, doping concentration) and produce specific outputs like electronic band structures, their computational expense often limits thorough parameter space exploration. This limitation drives the need for efficient surrogate models that can rapidly approximate DFT results while maintaining accuracy, thereby accelerating materials discovery.

## 8 CONCLUSION AND OUTLOOK

In this work, we presented RESuM, a rare event surrogate model designed for detector design optimization problems in physics. We began by statistically define the RED problem and proposed a CNP-enhanced surrogate model to solve it. We demonstrated the effectiveness of RESuM on optimizing the neutron moderator design for the LEGEND NLDBD experiment. Our results show that RESuM successfully identified an optimal design, reducing neutron background by 66.5% while utilizing only 3.3% of the computational resources required by traditional methods. The accuracy and coverage of the trained RESuM model were successfully validated with independently simulated HF datasets, showing that RESuM successfully emulate physical simulations with proper coverage and statistical robustness. Since all events (neutrons) are generated by traditional, first-principle-based physics simulation framework GEANT4, we believe that the RESuM model is more interpretable compared to other black-box surrogate models, such as those based on VAE or GANs (de Oliveira et al., 2017; Z. Fu et al., 2024). Although this work focuses on a specific detector design problem in physics, we believe that RED problems are prevalent in many other domains, as discussed in Section 7. Our future work will focus on two key directions: first, we want to further refine the RESuM model by validating with more HF simulations and improving the active learning algorithm; second, we intend to explore additional application of the RESuM model, especially in simulating Binary Black Hole mergers in Astronomy, as outlined in Section 7. This effort seeks to foster greater collaboration across the machine learning, physics, and astronomy communities, ultimately benefiting all fields involved.

## 9 ACKNOWLEDGEMENT

A. L. thanks the University of California, San Diego and San Diego Supercomputer Center for providing resources to this work. The work at Lawrence Berkeley National Laboratory (LBNL), including computational resources provided by the National Energy Research Scientific Computing Center (NERSC), is supported by the U.S. Department of Energy (DOE) under Federal Prime Agreement DE-AC02-05CH11231. The authors thank Yue Ma for her significant contribution to this work. The authors also thank Simon Mak, Yian Ma, the MODE (Machine Learning Optimized Design of Experiment) and LEGEND collaborations for valuable discussions.

## 10 REPRODUCIBILITY STATEMENT

To ensure that the RESuM model can be reliably reproduced, we have carefully documented all aspects of the methodology and experiment. The simulation tool and package used in our work is explicitly referenced in Section 5.1. The dataset, preprocessing steps, and model architecture are described in Section 4.2, Section 5.1 and Section 5.2. Model parameters and evaluation metrics are clearly defined in Section 4.2 and Section 5.2. The code of this work is anonymized and released as the supplementary material of this submission. All scripts for data handling, model training, and evaluation are included in the supplementary material, along with environment specifications and fixed random seeds to minimize variability. The training data of this work is too large as it involves in expensive simulations. The authors plan to release training data in the camera-ready version.

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

## APPENDIX

## 11   MULTI-FIDELITY GAUSSIAN PROCESS

Our implementation builds upon the architecture described in Ravi et al. (2024). For completeness, we provide a succinct overview of the essential components below.

In a Gaussian process (GP) a function $\hat{f}(\boldsymbol{\theta})$ is modeled as:

$$\hat{f}(\boldsymbol{\theta}) \sim \mathcal{GP}(\mu(\boldsymbol{\theta}), k(\boldsymbol{\theta}, \boldsymbol{\theta}'))$$

where $k(\boldsymbol{\theta}, \boldsymbol{\theta}')$ is the covariance function.

Given data $D_N = \{\boldsymbol{\Theta}, y\}$, the posterior mean and variance at a new point $\boldsymbol{\theta}_*$ are:

$$\mu_* = K(\boldsymbol{\theta}_*, \boldsymbol{\Theta})K(\boldsymbol{\Theta}, \boldsymbol{\Theta})^{-1}y$$

$$\sigma_*^2 = K(\boldsymbol{\theta}_*, \boldsymbol{\theta}_*) - K(\boldsymbol{\theta}_*, \boldsymbol{\Theta})K(\boldsymbol{\Theta}, \boldsymbol{\Theta})^{-1}K(\boldsymbol{\Theta}, \boldsymbol{\theta}_*)$$

where $K$ is constructed by evaluating the covariance function $k$ at the corresponding input points. This model represents the joint distribution of multiple fidelities as a multivariate GP with a specified covariance structure. The covariance matrix in this model includes both correlation terms between fidelities and discrepancy terms within fidelities.

We assume an additive relationship between the High and Low fidelities, so that the HF model $\hat{f}_H(\boldsymbol{\theta})$ could be expressed in terms of the LF model $\hat{f}_L(\boldsymbol{\theta})$ with a discrepancy $\delta(\boldsymbol{\theta})$:

$$\hat{f}_H(\boldsymbol{\theta}) = \rho \hat{f}_L(\boldsymbol{\theta}) + \delta(\boldsymbol{\theta})$$

where $\rho$ is a scaling factor and $\delta(\boldsymbol{\theta})$ is another random variable modeled as a GP. According to this relationship, the joint distribution of the two random variables $\hat{f}_L(\boldsymbol{\theta})$ and $\hat{f}_H(\boldsymbol{\theta})$ could be calculated as:

$$\begin{pmatrix} \hat{f}_L(\boldsymbol{\theta}) \\ \hat{f}_H(\boldsymbol{\theta}) \end{pmatrix} \sim \mathcal{N}\left(0, \begin{pmatrix} K_{LL} & \rho K_{LL} \\ \rho K_{LL} & \rho^2 K_{LL} + K_\delta \end{pmatrix}\right)$$

Here $K_{LL}$ and $K_\delta$ are the covariance of $\hat{f}_L(\boldsymbol{\theta})$ and $\delta(\boldsymbol{\theta})$. For more fidelity levels, the method recursively applies:

$$\hat{f}_{H_i}(\boldsymbol{\theta}) = \rho_i \hat{f}_{L_i}(\boldsymbol{\theta}) + \delta_i(\boldsymbol{\theta})$$

## 12    ACTIVE LEARNING STRATEGY

We use integrated variance reduction to quantify and minimize the expected posterior variance over the input space $\boldsymbol{\theta}$. The goal of integrated variance reduction is to minimize the total variance across the design space:

$$\mathcal{I}(\boldsymbol{\theta} = \int_\Theta \left[ \sigma^2_{\text{prior}}(\boldsymbol{\theta}') - \sigma^2_{\text{post}}(\boldsymbol{\theta}' \mid \boldsymbol{\theta}) \right] d\boldsymbol{\theta}',$$

where $\sigma^2_{\text{prior}}(\boldsymbol{\theta}')$ denotes the prior variance of the GP at point $\boldsymbol{\theta}'$ and $\sigma^2_{\text{post}}(\boldsymbol{\theta}' \mid \boldsymbol{\theta})$ is the posterior variance at $\boldsymbol{\theta}'$ given observation $\boldsymbol{\theta}$. The new sampling point $\boldsymbol{\theta}_{\text{new}}$ is selected by maximizing this integrated variance:

$$\boldsymbol{\theta}_{\text{new}} = \arg \max_{\boldsymbol{\theta}' \in \Theta} \mathcal{I}(\boldsymbol{\theta}')$$

In the case of a Gaussian Process (GP), the acquisition function $\mathcal{I}(\boldsymbol{\theta})$ defined by integrated variance reduction simplifies to the Lower Confidence Bound (LCB) acquisition function under the assumption that the variance reduction directly translates to balancing the predicted mean and uncertainty:

$$\mathcal{I}_{LCB}(\boldsymbol{\theta}) \approx \frac{1}{\#\text{samples}} \sum_i^{\#\text{samples}} \frac{k^2(\boldsymbol{\theta}_i, \boldsymbol{\theta})}{\sigma^2(\boldsymbol{\theta})}.$$

with $\sigma^2(\boldsymbol{\theta})$ representing the predictive variance at the observed point $\boldsymbol{\theta}$ and $k$ is the Radial Basis Function (RBF) kernel with $\boldsymbol{\theta}_i$ sampled randomly (Paleyes et al., 2019). Furthermore, we aim to optimize the acquisition function under parameter constraints—which limit the allowable values of the design parameters—using a gradient descent algorithm to locate the global maximum. These constraints ensure that the parameters remain within feasible ranges, reflecting the practical and structural requirements necessary for maintaining the integrity and functionality of the overall design. The constraints are incorporated into the acquisition function by adding a mulitiplicative factor nullifying the acquisition function's values when constraints are violated and thereby discouraging the algorithm from exploring infeasible regions (Gelbart et al., 2014).

## 13    CONDITIONAL NEURAL PROCESS

The core principle of the Conditional Neural Process (CNP) framework is to approximate arbitrary random processes using Gaussian sampling, where the mean $\mu$ and variance $\sigma$ are parameterized by neural networks. In this section, we show that the CNP score $\beta$ can be viewed as an estimate of $t(\boldsymbol{\theta}, \boldsymbol{\phi})$, with the Gaussian distribution representing the posterior of $t(\boldsymbol{\theta}, \boldsymbol{\phi})$. Furthermore, the RED problem can be aligned with the theoretical framework of the Variational Autoencoder (VAE), as detailed in Kingma (2013), with $t(\boldsymbol{\theta}, \boldsymbol{\phi})$ interpreted as the latent vector.

We begin by formulating the RED problem within a Bayesian framework: The training data consists of a finite set of $X_{ki}$ values generated through simulation, and the objective is to construct an encoder $q$ that approximates the posterior distribution of $t(\boldsymbol{\theta}, \boldsymbol{\phi})$, conditioned on the observed dataset $\{X_{ki}, \boldsymbol{\phi_{ki}}, \boldsymbol{\theta_k}\}$.

$$q(t(\boldsymbol{\theta}, \boldsymbol{\phi})|X_{ki}, \boldsymbol{\phi_{ki}}, \boldsymbol{\theta_k}) \approx p(t(\boldsymbol{\theta}, \boldsymbol{\phi})|X_{ki}, \boldsymbol{\phi_{ki}}, \boldsymbol{\theta_k}) \tag{8}$$

According to Bayes' theorem, the conditioned posterior in Eq. 8 could be calculated with the likelihood of the observed dataset and the prior of $t(\boldsymbol{\theta}, \boldsymbol{\phi})$:

$$p(t(\boldsymbol{\theta}, \boldsymbol{\phi})|X_{ki}, \boldsymbol{\phi_{ki}}, \boldsymbol{\theta_k}) \propto \mathcal{L}(X_{ki}|\boldsymbol{\phi_{ki}}, \boldsymbol{\theta_k}, t(\boldsymbol{\theta}, \boldsymbol{\phi}))p(t(\boldsymbol{\theta}, \boldsymbol{\phi})) \tag{9}$$

The prior $p(t(\boldsymbol{\theta}, \boldsymbol{\phi}))$ is conventionally set as a constant. Let $\mathcal{L}_k$ represents the $k$-th simulation, the combined likelihood of the full dataset is therefore:

$$\mathcal{L}(X_{ki}|\boldsymbol{\phi_{ki}}, \boldsymbol{\theta_k}, t(\boldsymbol{\theta}, \boldsymbol{\phi})) = \prod_{k=1}^{K} \mathcal{L}_k = \prod_{k=1}^{K} \prod_{i=1}^{N_k} \text{Bernoulli}(x = X_{ki}|p = t(\boldsymbol{\theta_k}, \boldsymbol{\phi_{ki}})) \tag{10}$$

Note that the ground truth of $t(\boldsymbol{\theta}, \boldsymbol{\phi})$ is unknown. In the Bayesian framework, we can only estimate it with a probability density function, which is $q(t(\boldsymbol{\theta}, \boldsymbol{\phi}))$. The estimation of the likelihood is therefore:

$$\mathcal{L}(X_{ki}|\boldsymbol{\phi_{ki}}, \boldsymbol{\theta_k}, q(t)) = \prod_{k=1}^{M} \prod_{i=1}^{N_k} \int \text{Bernoulli}(x = X_{ki}|p = t(\boldsymbol{\theta_k}, \boldsymbol{\phi_{ki}}))q(t(\boldsymbol{\theta}, \boldsymbol{\phi}))dt \qquad (11)$$

It is important to note that the quantity $t$ here is not a variable but a function, and the integration is performed in the Hilbert space. The problem, therefore, becomes to find the function $q^*(t(\boldsymbol{\theta}, \boldsymbol{\phi}))$ so that:

$$q^* = \arg \max_{q \in f(\boldsymbol{\theta}, \boldsymbol{\phi})} \mathcal{L}(X_{ki}|\boldsymbol{\phi_{ki}}, \boldsymbol{\theta_k}, q(t)) \qquad (12)$$

Here $f(\boldsymbol{\theta}, \boldsymbol{\phi})$ represents the set of arbitrary normalized functions of $(\boldsymbol{\theta}, \boldsymbol{\phi})$. While this optimization problem is mathematically solvable, the integration computation in Hilbert space is mathematically non-trivial and computationally expensive, rendering this solution impractical.

Then, here comes the CNP model, which simplifies and tackles this optimization problem in the following steps:

First, approximate $q$ with parameterized Gaussian, which is a natural choice if we regard the task as a statistical parameter estimation for $t(\boldsymbol{\theta}, \boldsymbol{\phi})$:

$$q_{NN}(t(\boldsymbol{\theta}, \boldsymbol{\phi})) = \mathcal{N}(\mu_{NN}(\boldsymbol{\theta}, \boldsymbol{\phi}, \boldsymbol{w}), \sigma_{NN}^2(\boldsymbol{\theta}, \boldsymbol{\phi}, \boldsymbol{w}))|_{X_{ki}, \boldsymbol{\phi_{ki}}, \boldsymbol{\theta_k}} \qquad (13)$$

Then, the neural network is trained to minimize the likelihood described in Eq. 11. Therefore, the CNP model actually performs a statistical estimation for $t(\boldsymbol{\theta}, \boldsymbol{\phi})$ by approximating the posterior $q$ with Gaussians.

If we consider $t(\boldsymbol{\theta}, \boldsymbol{\phi})$ as the latent vector in the VAE model, the pre-defined Bernoulli process acts as a probabilistic "decoder", which generates data given the latent vector.

With this framework, our task can be generally described as follows: Assume we have a predefined probabilistic decoder, $p(\boldsymbol{x}|\boldsymbol{z}, \boldsymbol{c})$, where $\boldsymbol{z}$ represents the latent vector and $\boldsymbol{c}$ is the condition. Additionally, we have a simulation informed by domain knowledge that generates data based on the nominal latent vector $\boldsymbol{z^*}$, though its exact value is unknown. Our objective is to develop a surrogate model that performs statistical estimation, represented as $q(\boldsymbol{z}|\boldsymbol{x})$, to infer the nominal latent vector from the simulated data $\boldsymbol{x}$. The posterior distribution $q(\boldsymbol{z}|\boldsymbol{x})$ serves the same role as the probabilistic encoder in a VAE model. We can then sample the latent vector $\boldsymbol{z}$ from $q(\boldsymbol{z}|\boldsymbol{x})$, which can subsequently be used to generate the "reconstructed" (surrogate) data $\boldsymbol{x'}$. The illustration of this architecture is shown in Figure 6.

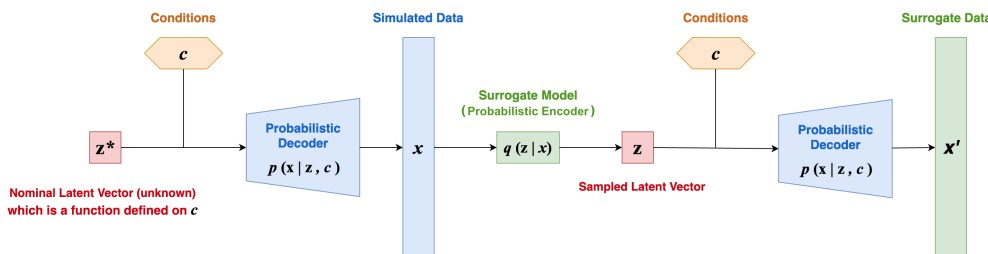

Figure 6: Architecture of Surrogate Model for the RED Problem

In this work, as the primary objective is to optimize the design parameters rather than generate additional simulated datasets, we utilize only the averaged CNP score $\beta$ as input to the subsequent MFGP model. Nevertheless, it is important to highlight that the trained CNP model has the potential to generate surrogate data as a fast simulator. This capability can be especially beneficial in other studies where the detailed distribution of the latent vector parameters is of interest, enabling more efficient exploration of the parameter space and supporting applications such as uncertainty quantification and model validation.

For the RED problem, specifically, $t(\boldsymbol{\theta}, \boldsymbol{\phi})$ corresponds to the latent vector $\boldsymbol{z}$, but with infinite dimensions, as it represents a continuous function over $(\boldsymbol{\theta}, \boldsymbol{\phi})$. $(\boldsymbol{\theta}, \boldsymbol{\phi})$ acts as the condition $\boldsymbol{c}$, and the fusion of latent vector and condition is performed by plugging $\boldsymbol{\theta}, \boldsymbol{\phi}$ into the function $t$ to get $t(\boldsymbol{\phi}, \boldsymbol{\theta})$

It is worth noting that the latent vector, denoted by $\boldsymbol{z}$, in our model, is not a conventional finite-dimensional vector but rather a vector situated in a Hilbert space. Dimensionality reduction can be achieved through quantization, which involves discretizing the space by creating a grid over $(\boldsymbol{\theta}, \boldsymbol{\phi})$ and computing values only at the selected grid points. An alternative approach is to project the latent vector onto a set of basis functions and impose a cutoff. For instance, with Fourier decomposition, the vector's coefficients can be retained up to a certain frequency limit, effectively serving as a high-frequency cutoff. In this work, we adopt the CNP model to represent it with a *fixed* dimensionality.

## 14 MODEL BENCHMARKING DETAILS

**Training and Validation Data** Additional independent, out-of-sample HF simulations are generated as validation dataset. Given the high computational demands of HF simulations and our limited resources, we generated 100 out-of-sample HF simulations at randomly sampled $\boldsymbol{\theta}$ values. The training dataset is identical to the RESuM training data described in Section 5, comprising 310 LF and 10 HF simulations. We further divided this dataset into three subsets: Small (305 LF + 5 HF), medium (307 LF + 7 HF), and full (310 LF + 10 HF). The small dataset establishes a threshold, meaning our RED problem at hand necessitates a minimum of 305 LF data samples to adequately capture the mapping of the 5-dimensional design space, as subsets with fewer samples would lack sufficient coverage. Additionally, some experiments reveal that at least 5 HF samples are required to establish a reliable correlation between LF and HF, enabling meaningful integration of the two fidelities.

**Baseline Models** There are two requirements for selecting baseline models: (1) the model must be multi-fidelity, and (2) it must be able to produce uncertainty estimates. Given these constraints, we selected two baseline models for benchmarking against the RESuM model (which combines MFGP and CNP): the Multi-Fidelity Gaussian Process (MFGP), and the Multi-Fidelity Bayesian Neural Network (MF-BNN). Leveraging several information sources, the MFGP and the RESuM frameworks employ a co-kriging approach such that the high-fidelity output is represented as: $y_{\text{hf}} = \rho \cdot y_{\text{lf}} + \delta$, where $\rho$ is a scaling factor and $\delta$ a discrepancy term -both inferred from the data. The MFGP and the RESuM model follow the architecture described in Chapter 11. In case of the Bayesian Neutral Network (BNN) (Bingham et al., 2019), we implemented a Multi-fidelity network by adopting a hierarchical architecture inspired by Guo et al. (2022). The MF-BNN integrates low- and high-fidelity data using two interconnected BNNs.

Each model, including RESuM, was trained on small, medium, and full training datasets and validated on an out-of-sample validation dataset. The benchmarking metrics, described in the next paragraph, were evaluated for each model. Additionally, to further demonstrate the robustness of the RESuM model, we performed 100 iterations, where each iteration involved randomly splitting the 410 LF and 110 HF samples into a full training dataset and a validation set (100 HF samples). The benchmarking metrics were then computed as averages across these 100 iterations to ensure statistical reliability and consistency.

**Benchmarking Metric** The ultimate goal of this manuscript is to learn the functional mapping $f : \boldsymbol{\theta} \to y$. Each trained model should be able to take $\boldsymbol{\theta}$ as input and predict $\hat{y}_{\text{Raw}}$ as the predicted and $\hat{\sigma}_{\text{Raw}}$ as the prediction uncertainty. Meanwhile, the ground truth $y_{\text{Raw}}$ for each HF simulation in the validation dataset is determined by counting the number of background events $m$. Based on these variables, we propose five validation metrics to assess the accuracy and coverage of each model:

- **MSE**: the mean square error $(\hat{y}_{Raw} - y_{Raw})^2$ summed over 100 HF validation datasets.
- **1/2/3$\sigma$[%]**: The 1/2/3$\sigma$ coverage of model prediction. This metric calculates the percentage of validation dataset for which $y_{\text{Raw}}$ falls within $\hat{y}_{Raw} \pm 1/2/3\hat{\sigma}_{\text{Raw}}$. Assuming a standard normal distribution, the expected coverage at $\pm 1\hat{\sigma}_{\text{Raw}}$, $\pm 2\hat{\sigma}_{\text{Raw}}$, and $\pm 3\hat{\sigma}_{\text{Raw}}$ are 68.27%, 95.45%, and 99.73% , respectively.
- **Cost**: The computational cost of generating the training data is shown as a percentage relative to the cost of traditional method (52,700 CPU hours). This metric depends only on

the size of the training dataset, with values of 1.1%, 2.3%, and 3.3% for the small, medium, and full datasets, respectively.

**Benchmarking Results** To provide a comprehensive understanding of the benchmarking results in Table 2, we include detailed plots in Figure 7 illustrating the statistical coverage for each baseline model for the full dataset (Trial 9, Trial 6, and Trial 3 of Table 2). The plots in Figure 7 visualize the percentage of ground truth $y_{\text{Raw}}$ (black marker) for all trials of the validation dataset that falls within the $1\hat{\sigma}_{\text{Raw}}$, $2\hat{\sigma}_{\text{Raw}}$, and $3\hat{\sigma}_{\text{Raw}}$ bands (shaded area) of the model prediction under study.

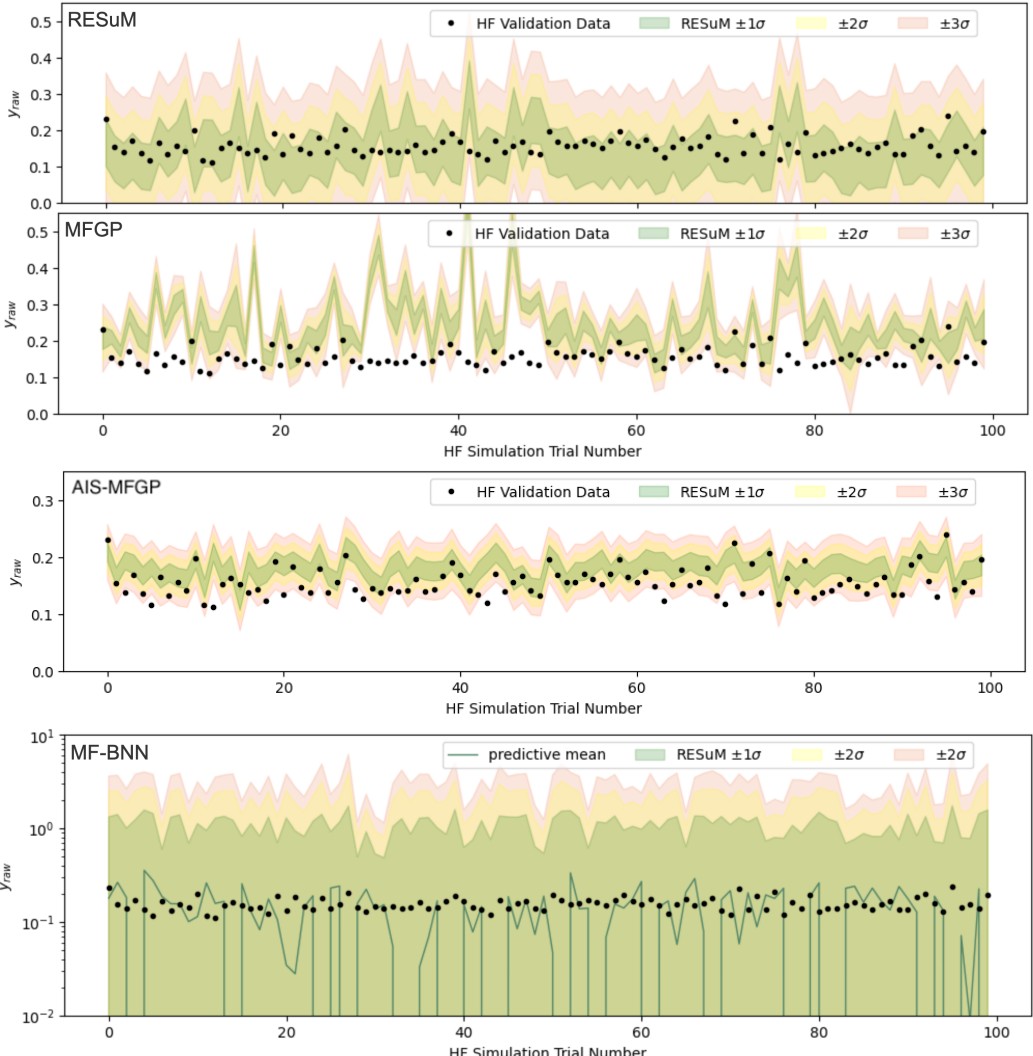

Figure 7: Validation comparison of RESuM (Upper) with the simplified MFGP (Upper Middle) AIS-MFGP (Lower Middle) and MF-BNN (Bottom) models using 310 LF and 10 HF data samples during training. Both comparison models fail to adequately describe $\hat{y}_{\text{Raw}}$, as demonstrated by their overly narrow (in case of MFGP) or wide (in case of MF-BNN) prediction bands and poor alignment with the HF validation data. Despite the narrow (wide) uncertainty bands, the predictions lack physical relevance, highlighting the models' inability to capture the complexity of the design space without the CNP model.

When comparing RESuM (Trial 12, Figure 7 (Upper)) to a simpler MFGP model (Trial 6, Figure 7 (Upper Middle) that only utilized the LF $y_{Raw}$ and HF $y_{Raw}$ of the exact same LF and HF data (excluding the CNP outputs $y_{CNP}$), the contrast in outcomes were striking. The simplified MFGP model not only failed to capture the complex dependencies between design parameters, but its predictions were also physically inconsistent. As shown in Figure 7 (Upper Middle), the prediction

bands for the simplified MFGP model are excessively narrow and do not capture the actual variability of the HF validation data. The model's inability to describe $\hat{y}_{\text{Raw}}$ is evident, as it does not adequately reflect the complex interactions within the design space. The model's $1\hat{\sigma}_{\text{Raw}}$, $2\hat{\sigma}_{\text{Raw}}$, and $3\hat{\sigma}_{\text{Raw}}$ confidence intervals are unrealistically tight, providing little insight into the true uncertainties of the system. With that, the coverage at $1\hat{\sigma}_{\text{Raw}}$, $2\hat{\sigma}_{\text{Raw}}$, and $3\hat{\sigma}_{\text{Raw}}$ or the simplified model was only 17%, 33% and 48%, a notably poor result compared to the RESuM model, which achieved a $1\hat{\sigma}_{\text{Raw}}$, $2\hat{\sigma}_{\text{Raw}}$, and $3\hat{\sigma}_{\text{Raw}}$ coverage of 69%, 95% and 100% with much more realistic uncertainty predictions. Similarly, the MF-BNN model (Trial 3, see Figure 7 Bottom) failed to adequately represent the data, resulting in predictions that lack physical interpretability and deviated from the system's expected behavior. The poor performance of the MF-BNN model can probably be attributed to the insufficient amount of training data, particularly the limited number of HF samples. The extremely wide uncertainty bands produced by the MF-BNN highlight the high degree of uncertainty in its predictions, further emphasizing its inability to effectively learn from the sparse training data. Meanwhile, an Adaptive Importance Sampling (AIS) approach combined with MFGP achieved $1\hat{\sigma}_{\text{Raw}}$, $2\hat{\sigma}_{\text{Raw}}$, and $3\hat{\sigma}_{\text{Raw}}$ coverage rates of 39%, 64%, and 86%, respectively. These results indicate that the AIS approach manages to sample from relevant regions and produce physically interpretable predictions, however, its coverage and overall performance remain significantly below RESuM's levels.

These findings underscore the complexity of the design space and the limitations of traditional Gaussian regression or neural network approaches with limited HF data. In contrast, the CNP-enabled RESuM model effectively reduced statistical variance and provided meaningful physical insights by capturing complex relationships between design parameters $\boldsymbol{\theta}$ and event-specific parameters $\boldsymbol{\phi}$.

# 15 ADAPTIVE IMPORTANCE SAMPLING

A major challenge in the Rare Event Detection (RED) problem is the limited number of signal events generated by low-fidelity (LF) simulations. To address this, we implemented Adaptive Importance Sampling (AIS), which strategically concentrates sampling efforts in regions more likely to produce signal events. In our standard LF simulations using uniform sampling, only 15-30 signal events occur per 50,000 neutron simulations.

We aim to conduct a benchmark comparison where we fed the AIS-enhanced LF simulations into the Multi-Fidelity Gaussian Process (MFGP) algorithm to compare its performance against the RESuM model. The AIS+MFGP benchmark can be considered as an ablation test of the CNP model. In this section, we focus on explaining the technical details of our AIS algorithm implementation.

Building on the rare event problem setting outlined in Section 3 and using the adaptive sampling framework, the expected probability $\bar{t}(\boldsymbol{\theta_k})$ of a signal event for a fixed design parameter $\boldsymbol{\theta_k}$ is calculated as the expectation value of $t(\boldsymbol{\theta_k}, \boldsymbol{\phi})$ over the target distribution $f(\boldsymbol{\phi} \mid \boldsymbol{\theta_k})$, where $f(\boldsymbol{\phi} \mid \boldsymbol{\theta_k})$ replaces $g(\boldsymbol{\phi})$ in Equation 2. It is closely related to $g(\boldsymbol{\phi} \mid \boldsymbol{\theta_k})$, the distribution of all event-specific parameters in the simulation. Specifically, $f(\boldsymbol{\phi} \mid \boldsymbol{\theta_k})$ can be expressed as:

$$f(\boldsymbol{\phi} \mid \boldsymbol{\theta_k}) = g(\boldsymbol{\phi} \mid \boldsymbol{\theta_k}) \cdot t_{\text{sig}}(\boldsymbol{\theta_k}, \boldsymbol{\phi}),$$

where $t_{\text{sig}}(\boldsymbol{\theta_k}, \boldsymbol{\phi})$ represents the probability of triggering a signal given the detector design $\boldsymbol{\theta_k}$ and the event-specific parameters $\boldsymbol{\phi}$. The trigger probability $t_{\text{sig}}(\boldsymbol{\theta_k})$, representing the overall probability of the Bernoulli process for signal events given $\boldsymbol{\theta_k}$, is defined as the expectation value of the indicator function $\mathbb{I}_{\text{sig}}(\boldsymbol{\phi})$, which evaluates to 1 if a signal event occurs and 0 otherwise. Mathematically, it is expressed as:

$$t_{\text{sig}}(\boldsymbol{\theta_k}) = \int \mathbb{I}_{\text{sig}}(\boldsymbol{\phi}) f(\boldsymbol{\phi} \mid \boldsymbol{\theta_k}) \, d\boldsymbol{\phi}.$$

This integral represents the density-weighted fraction of the parameter space that contributes to signal events. In practice, $\bar{t}(\boldsymbol{\theta_k})$ is approximated using Monte Carlo sampling as the weighted fraction of signal-producing samples from the target distribution. Specifically, adaptive sampling begins with a broad initial proposal distribution $q(\boldsymbol{\phi}; \boldsymbol{\theta_k}, 0)$, chosen to cover the entire event-specific parameter space $\boldsymbol{\Phi}$ for the given design $\boldsymbol{\theta_k}$. In each iteration $l$, samples $\boldsymbol{\phi}_i$ are drawn from the current proposal distribution $q(\boldsymbol{\phi}; \boldsymbol{\theta_k}, l)$. These samples are evaluated under $f(\boldsymbol{\phi} \mid \boldsymbol{\theta_k})$ to compute importance weights:

$$w_i = \frac{f(\boldsymbol{\phi}_i \mid \boldsymbol{\theta_k})}{q(\boldsymbol{\phi}_i; \boldsymbol{\theta_k}, l)}.$$

The proposal distribution is updated iteratively to better approximate $f(\phi \mid \boldsymbol{\theta_k})$. For Gaussian proposals, the weighted samples are used to recompute the mean and covariance matrix:

$$\boldsymbol{\mu}_{l+1} = \frac{\sum_{i=1}^{N_l} w_i \boldsymbol{\phi}_i}{\sum_{i=1}^{N_l} w_i}, \quad \boldsymbol{\Sigma}_{l+1} = \frac{\sum_{i=1}^{N_l} w_i (\boldsymbol{\phi}_i - \boldsymbol{\mu}_{l+1})(\boldsymbol{\phi}_i - \boldsymbol{\mu}_{l+1})^\top}{\sum_{i=1}^{N_l} w_i}.$$

This iterative process continues until $q(\boldsymbol{\phi}; \boldsymbol{\theta_k}, l)$ sufficiently concentrates on the high-probability signal regions in $\phi$ for the fixed design $\boldsymbol{\theta_k}$. By iteratively refining the sampling distribution, adaptive sampling reduces variance in the estimation of $\bar{t}(\boldsymbol{\theta_k})$.

We apply the Adaptive Importance Sampling (AIS) framework to each LF simulation with a fixed design parameter $\boldsymbol{\theta_k}$, allowing efficient approximation of the conditional signal distribution $f(\boldsymbol{\phi} \mid \boldsymbol{\theta_k})$ and increasing neutron capture occurrences. Initially, a proposal distribution $q(\boldsymbol{\phi}; \boldsymbol{\theta_k}, 0)$ is constructed using k-means clustering on event-specific input parameters (d = 2), such as position, momentum, and energy. Clusters emphasize regions where the moderator strongly influences neutron interactions. In each iteration $l$, the AIS algorithm draws samples $\boldsymbol{\phi}_i$ from the current proposal and assigns importance weights $w_i$, reflecting the ratio of the target density to the proposal density. These weighted samples drive updates to the Gaussian mixture components of $q$, recalculating their means and covariances to better approximate $f(\boldsymbol{\phi} \mid \boldsymbol{\theta_k})$. Covariance matrices are regularly checked and "regularized" for numerical stability, while an optional Principal Component Analysis (PCA) step may reduce dimensionality to further stabilize updates. By iteratively adding and adjusting Gaussian components based on where weighted samples cluster, the proposal distribution adapts toward regions deemed most significant by the target density.

In each iteration, once the samples are drawn and their importance weights are computed, a simulation run is performed with these sampled points and the results are added to the existing sample pool, forming an aggregated distribution. This aggregated distribution is then included in the denominator of the weight calculation. This helps maintain broader coverage, prevents mode collapse, and leverages prior knowledge. As the algorithm progresses over multiple iterations, this aggregation acts as a "safety net" distribution, maintaining exploration in under-sampled regions while still allowing the main proposal components to adapt toward regions of high target density. Diagnostics such as Effective Sample Size (ESS) are monitored to gauge performance and convergence, with higher ESS indicating that the proposal distribution is capturing the target distribution more effectively.

Challenges arise from the limited signal-to-background ratio $(2 : 10^3)$ in LF simulations, which impedes identification of statistically meaningful clusters. To ensure statistically meaningful covariance matrix estimation, a heuristic suggests that each cluster must contain at least $d(d+1)$ samples (Chen et al., 2010). The initial proposal is derived from 40% of the available event samples for the RESuM analysis. Using 2 clusters and with fewer than 15 signal events on average per initial proposal, this heuristic is not satisfied in early stages, as shown in Figure 8. Nevertheless, we purposely do not merge data across different LF simulations for the AIS k-means clustering because each simulation uses a distinct, fixed design parameter $\theta$. The merges of all data shown in Figure 8 are for demonstration purposes only. For evaluation of the quality of the clustering, we show the silhouette score, which measures how similar a data point is to its assigned cluster compared to other clusters. A higher silhouette score indicates better-defined and well-separated clusters. The incorporated constraints focus sampling on regions of relevance while maintaining well-separated clusters with a silhouette score (Rousseeuw, 1987) exceeding 0.5 (see Figure 8 Bottom). This balance between exploration and exploitation ensures robust initialization and refinement of the AIS proposal distribution.

The AIS framework in combination with a MFGP achieved coverage rates of 39%, 64%, and 86% for $1\hat{\sigma}_{\mathrm{Raw}}$, $2\hat{\sigma}_{\mathrm{Raw}}$, and $3\hat{\sigma}_{\mathrm{Raw}}$, respectively. While the AIS approach has demonstrated its ability to improve the efficiency of LF simulations by increasing the signal-to-background ratio and directing sampling toward relevant regions, it still does not achieve the performance level of RESuM within the same 50,000-sample budget (see Figure 7 (Lower Middle)); additional samples would be required to ensure more robust coverage and convergence.

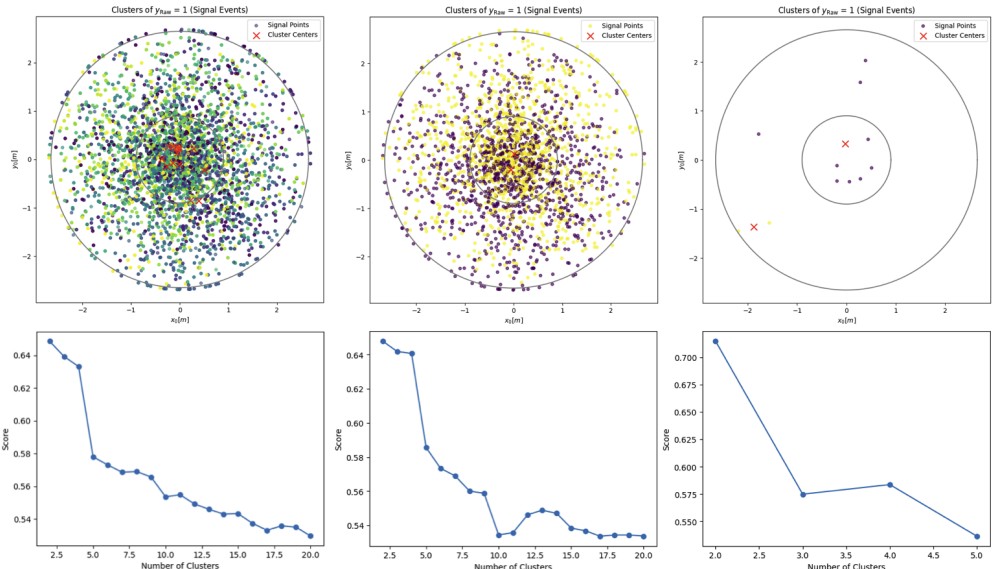

Figure 8: Visualization of signal samples and K-means cluster centers (top row) and corresponding silhouette scores (bottom row) for different data conditions. The first column shows the clustering of all signal samples using 100% of the available data, while the second column shows clustering with only 40% of the available data. The third column shows the signal distributions of one LF simulation using 40% of the available event samples. The bottom row illustrates the silhouette score as a function of the number of clusters for each corresponding scenario, highlighting the clustering quality under different data availability and conditions.

