# OpenReview forum: "RESuM: A Rare Event Surrogate Model for  Physics Detector Design"
_ICLR.cc/2025/Conference — ICLR 2025 Spotlight_

### Official Review · Reviewer_TJgd · 2024-11-01

**Soundness:** 3
**Presentation:** 3
**Contribution:** 3
**Rating:** 6
**Confidence:** 2

**Summary:**

The paper introduces RESuM (Rare Event Surrogate Model), designed to optimize physics detector design, specifically for reducing background events in neutrinoless double-beta decay (NLDBD) detection. The authors frame the challenge as a Rare Event Design (RED) problem, where background events are so rare that traditional simulation methods become computationally prohibitive. RESuM employs a multi-fidelity approach, using a Conditional Neural Process (CNP) model to learn from limited data and a Gaussian Process (GP) model to efficiently explore the design space. The authors apply RESuM to the LEGEND experiment’s neutron moderator design and achieve an  efficient performance.

**Strengths:**

1. The model targets a critical challenge in physic.
2. The integration of CNP and MFGP is a novel approach for handling rare event problems.
3. The framework’s formulation makes it generalizable to other simulation-heavy domains.

**Weaknesses:**

I do not find major concerns.
Minor issues:
1. The success of RESuM’s multi-fidelity modeling approach relies on access to both high-fidelity and low-fidelity simulation data. In situations where these data sources are unavailable or highly dissimilar, would the model's effectiveness be reduced?
2. It seems to me that the RESuM does not embed physics constraints or validation check to the model, thus it might be possible that RESuM would suggest some invalid designs. In practice, do we need to first define the valid range of parameters for RESuM? And thus for different application contexts, we need to modify the model accordingly?
3. Section 5 uses subsubsections without subsections.
4. Can RESuM scale to applications with larger or more complex detector designs? How to estimate the relationship between the dimension of design space and model performance? Also, can RESuM incorporate expert knowledge on the design space to make it more efficient?

**Questions:**

See Weaknesses for questions.

---

> ### Author Response · Authors · 2024-11-21
>
> We'd like to thank the reviewer's comments and questions. We highly appreciate the insightful questions reviewer asked and think those these questions point to the key aspects of our model. Here are our tailored responses to those questions:
>
> > The success of RESuM’s multi-fielity modeling approach relies on access to both high-fidelity and low-fidelity simulation data. In situations where these data sources are unavailable or highly dissimilar, would the model's effectiveness be reduced?
>
> Response: Thanks for the comment, this is a very good point. Indeed one of the prerequisite for RESuM is the access of both HF and LF simulation data, and the major motivation is to use LF data as a cheap/quick proxy to scan a larger design space. In our study, the HF/LF data are mainly different in two aspects:
>
> * number of simulated events: LF simulations typically include on the order of  10^4  neutrons, while HF simulations involve 10^7  muons, approximately 30% of which create neutrons. The smaller number of events in the LF simulations introduces higher statistical noise, which the RESuM model accounts for in its modeling process.
> * amount of physics information incorporated in each event: LF neutron events are generated from a simplified uniform distribution, whereas HF data is derived from the actual muon flux, providing a more realistic representation.
>
> If the difference between HF/LF simulation takes another form, then the effectiveness of RESuM model might be impacted. This is the future research direction we would like to explore: generalizing RESuM onto other applications where LF/HF simulations might have other kind of differences. For example, difference in granularity (HF: simulated on fine grid; LF: simulated on coarse grid).
>
> > It seems to me that the RESuM does not embed physics constraints or validation check to the model, thus it might be possible that RESuM would suggest some invalid designs. In practice, do we need to first define the valid range of parameters for RESuM? And thus for different application contexts, we need to modify the model accordingly?
>
> Response: Thank you for this excellent comment! RESuM does incorporate physics constraints and checks to make sure the suggested designs are valid. In our study, the constraints include parameter ranges and correlated conditions, such as the inner and outer radius of the panels, and other geometric relationships.These constraints are built into both the initial sampling of the design parameters for the training data and into the active learning process.
> * Initial sampling: As it can be seen in Figure 3, the sampled data are not uniformly distributed within the parameter ranges, reflecting these intercorrelated constraints we can express mathematically.
> * Active learning strategy: As mentioned in Section 10 (Section 11 in the updated manuscript), if a design violates these constraints, a multiplicative factor in the acquisition function becomes zero, preventing the model from exploring those invalid regions. We have added additional details regarding this in the new manuscript.
>
> While different applications may require updated user knowledge, RESuM is designed to handle these changes as long as they can be expressed in analytical form. Incorporating these user inputs is done in the following order: first, the user defines the parameter ranges, and then any additional constraints are formulated in an analytical form to ensure they can be properly incorporated into the model.
>
> > Section 5 uses subsubsections without subsections.
>
> Response: Thanks for pointing this out, we have fixed it in the new manuscript.

---

> > ### Author Response · Authors · 2024-11-21
> > **Response to last questions**
> >
> > > Can RESuM scale to applications with larger or more complex detector designs? How to estimate the relationship between the dimension of design space and model performance? Also, can RESuM incorporate expert knowledge on the design space to make it more efficient?
> >
> > * Thanks for the question. One of the distinction between RESuM and other surrogate model is that its lowest level data point are always generated by traditional physics simulation based on first principles (in our case, each neutron is simulated by the GEANT4 simulation software). For larger or more complex detector such as LHC, such information can be generated by GEANT4 simulation, then we believe RESuM could be applied as well.
> > * In our case, RESuM deals with a 5D design space and 1D design metric, which is relatively simple. We’d like to point out that the 5 dimensions are not equally important in influencing the design metric (for example, Figure 4 shows that Radius is important while Thickness affects y less strongly). We believe that RESuM has the capability to learn these relationship thereby “ignoring” less important dimensions. However, one should also note that GPs might face challenges with the curse of dimensionality and sufficient training data for complex problems will be needed to cover the expanded space.
> > * Some level of expert knowledge has already been incorporated in the active learning part, as we discussed in the previous questions. Additional expert knowledge can be incorporated if we can output multiple design metric into active learning (i.e. surrogating both the neutron reduction rate and cost of neutron moderator, optimizing its performance while minimizing the cost).

---

> > > ### Comment · Reviewer_TJgd · 2024-11-22
> > >
> > > Thank you for addressing my concerns in the rebuttal. The revisions effectively resolved the issues I raised and provided additional details on the experiments, including comparisons to baselines. These updates are more convincing and strengthen the paper's contributions. I will maintain my positive score.

---

> > > > ### Author Response · Authors · 2024-11-24
> > > >
> > > > Thank you for your prompt response and continued support of our paper's acceptance. On behalf of all authors, we found your questions to be very insightful and valuable, and they have inspired our thinking for how to improve the RESuM model in the future.

---

### Official Review · Reviewer_KsYG · 2024-11-02

**Soundness:** 3
**Presentation:** 3
**Contribution:** 2
**Rating:** 5
**Confidence:** 3

**Summary:**

This work presents a generative approach for optimizing physics detector design in rare event scenarios by leveraging a Conditional Neural Process (CNP). By incorporating techniques like simple co-kriging, data augmentation, and multi-fidelity modeling, the authors demonstrate that their surrogate model effectively addresses the computational demands of rare event detection while maintaining the accuracy.

**Strengths:**

The authors provide a solid background introduction to the NLDBD problem and offer clear, detailed explanations of the technical methods used in this work, including co-kriging and Conditional Neural Processes (CNP), in the appendix.

**Weaknesses:**

Some of the critical weaknesses:

- While the authors provide a solid introduction to NLDBD and its motivation, the paper lacks a literature review on rare event simulation/modeling. Rare event simulation is a well-established field in engineering, including techniques such as the first/second-order reliability methods (FORM/SORM) [1, 2, 3], polynomial-based surrogate modeling [4, 5], adaptive and sequential importance sampling [6, 7], and ensemble Kalman filters [8]. These approaches have also been extended to multi-fidelity settings, as seen in [9, 10, 11]. Although it’s not necessary to cite all related works, the authors should clarify why they focus on deep generative modeling, highlighting its advantages over traditional models.

- The computational results of the proposed method appear to lack a baseline, making it challenging to assess performance without a fair comparison.

- Some notations are not statistically rigorous. Specific comments on notations are provided in the questions section. Also, the variables $\phi$ and $\theta$ are inconsistently bolded, which can cause confusion.

Overall, we find that this work falls below the acceptance threshold for ICLR. We encourage the authors to: (1) expand the literature review to include rare event simulation/probability estimation as outlined here, (2) refine the statistical notations, and (3) add baselines for experiments. With these adjustments, this work would be in a better shape.

[1] A. M. Hasofer, An exact and invarient first order reliability format, J. Eng. Mech. Div., Proc. ASCE 100 (1974) 111–121.

[2] A. Der Kiureghian, et al., First-and second-order reliability methods, Engineering design reliability handbook 14 (2005).

[3] B. Fiessler, H.-J. Neumann, R. Rackwitz, Quadratic limit states in structural reliability, Journal of the Engineering Mechanics Division 105 (1979) 661–676.

[4] J. Li, D. Xiu, Evaluation of failure probability via surrogate models, Journal of Computational Physics 229 (2010) 8966–8980.

[5] J. Li, J. Li, D. Xiu, An efficient surrogate-based method for computing rare failure probability, Journal of Computational Physics 230 (2011) 8683–8697.

[6] C. G. Bucher, Adaptive sampling—an iterative fast monte carlo procedure, Structural safety 5 (1988) 119–126.

[7] I. Papaioannou, C. Papadimitriou, D. Straub, Sequential importance sampling for structural reliability analysis, Structural safety 62 (2016) 66–75.

[8] F. Wagner, I. Papaioannou, E. Ullmann, The ensemble kalman filter for rare event estimation, SIAM/ASA Journal on Uncertainty Quantification 10 (2022) 317–349.

[9] B. Peherstorfer, T. Cui, Y. Marzouk, K. Willcox, Multifidelity importance sampling, Computer Methods in Applied Mechanics and Engineering 300 (2016) 490–509.

[10] B. Peherstorfer, B. Kramer, K. Willcox, Multifidelity preconditioning of the crossentropy method for rare event simulation and failure probability estimation, SIAM/ASA Journal on Uncertainty Quantification 6 (2018) 737–761.

[11] F. Wagner, J. Latz, I. Papaioannou, E. Ullmann, Multilevel sequential importance sampling for rare event estimation, SIAM Journal on Scientific Computing 42 (2020) A2062–A2087.

**Questions:**

- This works uses CNP as the major tool for rare event modeling. Is it model originally proposed by the authors? If not, please provide the corresponding citations. Also, please give the motivation here. Why CNP is more advantageous than VAE/GAN in this setting?

- In line 51, the authors said "If $m_1/N_1 < m_2/N_2$, it suggests that the design $\theta_1$ is better than $\theta_2$." This statement holds only if $N_1, N_2$ are sufficiently large, right? Otherwise, since it is in a rare event scenario, a bad $\theta$ may also lead to zero values when $N$ is not large enough.

- It looks like $X_{ki}$ is equivalent with $t(\theta_k, \phi_{ki})$, why use two separate notations? Please correct me if I am wrong.

- Formula (4): I don't understand what does Bernoulli($p=t(\theta_k, \phi_{ki})$) mean here. Do you mean Bernoulli($p=\bar{t}(\theta_k)$)? Also, the letter $p$ has been used as density function.

- The variable $\beta$ is applied in different versions without clear definitions. In line 160, the authors introduce $\beta_{ki}$, while $\beta$ appears in formula (5) without clearly showing its dependency. My guess is that $\beta_{ki} = \beta(\theta_k,\phi_{ki})=\beta\vert\theta_k, \phi_{ki}=\beta$. Please clarify.

- Line 182, typo: "RESuM as shown in Figure 3." Here it is supposed to be Figure 1.

- Line 230: you don't need to give the fourmulation of the acquisition function, since $\boldsymbol{x}$ is not introduced here.

- Line 299 and line 308: The density $g(\phi)$ for LF and HF models are different, right? If so, please use different notations for them.

- Line 517-520: "Based on the statistical formulation and ... for accelerating simulations. " I don't see the comparison between the proposed method with VAE/GANs, could you please clarify how you draw this conclusion?

---

> ### Author Response · Authors · 2024-11-21
> **Clarifying the difference between RED problem and rare event simulation/modeling**
>
> We thank the reviewer for their thorough review and valuable references to the rare event simulation and probability estimation literature. We have cited all 11 suggested citations in our revised manuscript. While our background is in physical sciences rather than engineering, we have carefully studied the references and conducted additional literature reviews to understand rare event simulation and probability estimation.We would like to respectfully point out the distinctions between our work and rare event simulation approaches. The key differences are listed below:
> * The task in rare event simulation and probability estimation is to calculate this integral with **known performance function G(x) and probability density function f(x)**: $P_{f}=\mathrm{Prob}[G(X)\leq0]=\int_{G(X)\leq 0}f(X)dX$. In this context, surrogate model means using FORM/SORM, polynomial-based surrogate methods or other methods to approximate G(x), while using adaptive importance sampling to sample from f(X) to accelerate the calculation.
> * The RED problem differs from the rare event simulation problem in that: (1) In our case, the function G(X) of interest is a joint probability distribution t(θ,φ) (2) Unlike G(x) which is known, **t(θ,φ) is not known**, we only have limited access to its outcome Bernoulli[t(θ,φ)], which is either 0 (not triggered) or 1 (triggered).
>
>   * This motivates the use of CNP: since Bernoulli[t(θ,φ)] can only be either 0 and 1, when we count the total number of events triggering the signal m, the design metric y=m/N becomes highly stochastic especially when N is small. However, CNP is able to accept θ,φ and predict a floating point number between 0 and 1, where 0.8 could mean this event is 80% likely to create a signal. Our work has shown that this efficiently smooths out the discreteness in y.
>
> * (3) As Reviewer BhVu pointed out, RESuM serves as a surrogate that learns a mapping between θ and y over the entire design space θ, encompassing both good and bad designs. We used Bayesian methods such as the Gaussian Process, which learn the mapping between the input space X and probability distributions over possible output values. This differs from rare event simulation, which employs adaptive sampling or iterative methods to sample more in regions where G(X) < 0.
>
> Given these discrepancies, we believe that the RED problem we aim to solve in this work is different from the rare event simulation/modeling highlighted by the reviewer. We summarized these distinctions in Section 2: Related Works in the revised manuscript.
>
> > add baselines for experiments
>
> Response: Thank you for your comment. For a detailed discussion of our new benchmarking study, we respectfully refer you to the general comment and updated manuscript.
>
> > Notations are not statistically rigorous
>
> Response: Thank you for your comment. We have modified our statistical notation. The bolded symbol of theta and phi means it is a vector instead of a scalar. We have gone through our paper again and bolded all the theta and phi’s to keep consistency.

---

> ### Author Response · Authors · 2024-11-21
> **Response to Reviewer's Questions**
>
> > This works uses CNP as the major tool for rare event modeling. Is it model originally proposed by the authors? If not, please provide the corresponding citations. Also, please give the motivation here. Why CNP is more advantageous than VAE/GAN in this setting?
>
> Response:
> * Thanks for the question. The motivation for CNP was discussed in our previous response. We cited CNP in line 141 of the original draft when discussing rare event surrogate models. Following your suggestion, we have moved the CNP citation to our introduction.
> * While VAE/GAN are generative models, CNP serves a different purpose in the RESuM model: it smooths out discrete simulation outputs by approximating  t(θ, ϕ)  with a CNP output score beta. It accepts theta and phi as input and output a single, floating point score beta. Beta can be considered as a continuous probability between 0 and 1. For example, if beta = 0.8 indicates an 80% probability of signal triggering for that event. Since CNP functions as a predictive model rather than a generative model, direct comparison with VAE/GAN architectures would not be possible.
>
> > In line 51, the authors said "If $m_1/N_1 < m_2/N_2$, it suggests that the design $\theta_1$ is better than $\theta_2$." This statement holds only if $N_1, N_2$ are sufficiently large, right? Otherwise, since it is in a rare event scenario, a bad $\theta$ may also lead to zero values when $N$ is not large enough.
>
> Response: Thanks for pointing this out. This is the key challenge we would like to address in the RED problem: when N is small, the design metric y may lead to 0 as the reviewer pointed out, or subject to significant statistical variation; when N is large, the simulation become prohibitively expensive. We discussed this in details in the “Small N Scenario vs. Large N Scenario” in Section 3, and the RESuM model aims to surrogate this design metric given that we only have access to a limited amount of N.
>
> > Line 299 and line 308: The density for LF and HF models are different, right? If so, please use different notations for them.
>
> Response: Thank you for your comment, we have changed the LF density to $g_n^{ini}(x)$.
>
> >  It looks like $X_{ki}$ is equivalent with $t(\theta_k, \phi_{ki})$, why use two separate notations? Please correct me if I am wrong.
>
> Response: Thanks for the question. $X_{ki}$ and $t(\theta_k, \phi_{ki})$ are not the same. In fact, $X_{ki} \sim Bernoulli[t(\theta_k, \phi_{ki})] $, so $X_{ki}$ is binary but $t(\theta_k, \phi_{ki})$ is continuous. In RED problem, We only have access to $X_{ki}$ but we do not know $t(\theta_k, \phi_{ki})$. We updated equation 4 to reflect this.
>
> > Formula (4): I don't understand what does Bernoulli($p=t(\theta_k, \phi_{ki})$) mean here. Do you mean Bernoulli($p=\bar{t}(\theta_k)$)? Also, the letter $p$ has been used as density function.
>
> Response: Thanks for the comment. We realize this equation is a bit confusing. Our original intention was: “$X_{ki}$ follow a probability distribution p($X_{ki}|\theta, \phi$) which is a bernoulli distribution of $t(\theta_k, \phi_{ki})$”. We rewrote Equation (4) as $X_{ki} \sim \text{Bernoulli}(t(\boldsymbol{\theta_k},\boldsymbol{\phi_{ki}}))$ and we deleted the “p=” in the Bernoulli().
>
> > The variable $\beta$ is applied in different versions without clear definitions. In line 160, the authors introduce $\beta_{ki}$, while $\beta$ appears in formula (5) without clearly showing its dependency. My guess is that $\beta_{ki} = \beta(\theta_k,\phi_{ki})=\beta\vert\theta_k, \phi_{ki}=\beta$. Please clarify.
>
> Response: Thanks the comment, this is correct. $\beta$ indicates CNP score in general, while $\beta_{ki}$ indicates the CNP score of a specific event: the ith event in the kth simulation trial. We have clarified this after equation 5 and 6.
>
> > Line 182, typo: "RESuM as shown in Figure 3." Here it is supposed to be Figure 1.
>
> Response: Thanks, this has been corrected.
>
> > Line 517-520: "Based on the statistical formulation and ... for accelerating simulations. " I don't see the comparison between the proposed method and VAE/GANs. Could you please clarify how you drew this conclusion?
>
> Response: Thanks for the question. In this work, all events(neutrons) are simulated using traditional physics simulation software (GEANT4), which is both fully interpretable and grounded in well-established physical first principles. None of the events are generated by VAE/GAN or other generative models, which typically operate as black-box. Instead, the RESuM model's strength lies in minimizing the number of simulations needed while achieving both accurate predictions and proper statistical coverage with limited data.

---

> > ### Comment · Reviewer_KsYG · 2024-11-23
> >
> > We greatly appreciate the authors' added experiments and detailed responses to our reviews. Most of our concerns regarding the notations and the experiments have been resolved. We also acknowledge the authors' efforts to incorporate our suggested citations into the draft. However, we respectfully disagree with the authors' claim:
> >
> > > "The RED problem differs fundamentally from traditional reliability engineering rare event problems, as it must estimate failure probabilities (triggering probabilities in RED) without an explicitly known performance function G(x). (Updated draft line 92)"
> >
> > We would like to provide the following clarifications:
> >
> > - The performance function $G(X)$ is not necessarily known in prior methods. For example, in the FORM/SORM and polynomial surrogate model approaches, the limit state function (not the performance function) $\{x: G(x) = 0\}$ is approximated using various surrogate models. These approaches assume the continuity of $G(x)$ but do not require it to be explicitly defined. For sampling-based methods, only access to the indicator results $\mathbb{1}_{G(x)<0}$ is required, which is equivalent to $X$ drawn from $\text{Bernoulli}(t(\boldsymbol{\theta}, \boldsymbol{\phi}))$, thereby sharing the same setting as this work.
> >
> > - Building on this point, the problem in this work's setting can be addressed using existing (importance) sampling-based methods. For any given $\boldsymbol{\theta}_k$, the corresponding $\bar{t}(\boldsymbol{\theta}_k)$ can be precisely estimated by applying these methods to compute $P_f(\boldsymbol{\theta}_k)$. Subsequently, the optimal parameter $\boldsymbol{\theta}^\ast$ can be identified by minimizing $P_f(\boldsymbol{\theta})$. These established methods offer solid convergence guarantees and have been validated across numerous applications. Therefore, we strongly believe at least one of these approaches should be included as a competing method for performance comparison. The absence of such reliability engineering-based methods significantly reduces the novelty of this work.
> >
> > We hope the authors will consider these points for further improving the quality of the manuscript.

---

> > > ### Author Response · Authors · 2024-11-25
> > >
> > > We thank the reviewer for the prompt reply. The reviewer is correct that advancements in FORM/SORM and IS have addressed some of their traditional limitations, particularly with surrogate models and indicator-based sampling. We understand the reviewer’s concern regarding the absence of a direct comparison to traditional approaches commonly used in rare event simulations. We are currently developing a combined Importance Sampling and Gaussian Process (IS+GP) model as a baseline for comparison with RESuM. While we aim to show the benchmarking results by December 2nd (the extended deadline), we cannot guarantee this delivery due to two factors:
> > > * the technical challenges we encountered in model development which we detail in subsequent comments
> > > * the availability of our computational cluster resources
> > >
> > > Nevertheless, we would like to highlight both the challenges we encountered in developing the IS+GP model, as well as how the proposed CNP/RESuM model in this paper addressed these challenges:
> > > 1. __Complex, discontinue G(x) space:__ Surrogate-enhanced FORM/SORM and IS can struggle with discontinuous settings or irregular problem boundaries as they rely on smoothness in G(X). As [Zhou et al., 2018] demonstrates, surrogate-enhanced FORM/SORM methods can perform poorly when applied to problems with sharp transitions or fragmented G(X) domains, when the underlying surrogates struggle to adequately capture non-smooth characteristics. This problem becomes particularly severe when we want to model the entire design space G(X) rather than only sampling in a specific, smooth region.
> > >    * In contrast RESuM leverages the CNP as a novel solution for a flexible estimation of G(X) by probabilistic inference. This allows us to capture both local irregularities and global patterns in G(X). The CNP can provide reliable estimates of G(X) even in scenarios where data is sparse, highly variable, or discontinuous.
> > >
> > >
> > > 2. __Integrating IS with Field-Standard GEANT4 Simulation:__  this paper aims to suggest detector design for NLDBD experiment. The field of NLDBD search has been established for over 70 years and there is a sophisticated, well-recognized simulation framework (GEANT4) adopted and trusted by the majority of the field. To propose credible detector designs, we are required to conduct our simulations in GEANT4—following the established protocols and validation methods agreed upon by the physics community.
> > >    * GEANT4 is already making use of biasing methods [for details see the GEANT4 Application Developer Guide Section 3.7]. And we already include a GEANT4’s internal biasing technique for the neutron’s cross-sections boosting the neutron production when generating HF simulations. However, there are studies highlighting discrepancies between experimental measurements and the sampling outcome of these methods, particularly for neutrons when biasing weights are set too high [Ziagokova et al. 2023]. This paper's observation agrees with Point 1.
> > >    * On the contrary, RESuM directly trains on data generated by GEANT4, without modifying any of its existing structure/sampling strategy. This makes RESuM complies with the established field standard in NLDBD search, making it easier for physicist to understand and accept its suggested detector design.
> > >    * IS requires iterative sampling adjustments [Owen and Zhou (2000),  Arazo et al. (2021), Morzfeld et al. (2016)], which has to be customized and carefully integrated into the GEANT4 pipeline. We have attempted this integration in the last few days and it is highly non-trivial in the current existing pipeline of our simulation study. While we believe enhancing GEANT4 with advanced IS methods is a promising future project, such development effort is not one or two days of work and it extends beyond the scope of the current RESuM paper.
> > > 3. __Uncertainty and Scalability:__ Traditional methods like IS and FORM/SORM are well-established for specific rare-event estimation but remain computationally expensive for high-dimensional or global problems [Bengtsson, Bickel, and Li (2005),Li, H.,Der Kiureghian and Ditlevsen (2009), Mahadevan, S. (2022)]. RESuM’s surrogate-based approach drastically reduces the computational cost and sampling burden while providing broader, uncertainty-aware predictions, making it more efficient for optimization and exploration tasks.

---

> > > > ### Comment · Reviewer_KsYG · 2024-11-26
> > > >
> > > > Thank you for acknowledging the challenges in implementation. We understand the difficulty of adding additional results under a tight deadline and appreciate your discussions on the comparison between FORM/SORM and CNP. The idea of combining IS with GP sounds very interesting, and we would be curious to see how it performs in future work.
> > > >
> > > > We believe the current draft is in a better shape; however, we still think a comparison with traditional methods, such as IS, is essential. We have raised the scores accordingly to reflect our evaluation of the manuscript's current state.

---

> ### Author Response · Authors · 2024-12-02
>
> We thank the reviewer for their valuable feedback and increased score. We have dedicated substantial time in designing an Adaptive Importance Sampling (AIS) algorithm for our data. While we recognize AIS's potential to yield better results, our current implementation represents our best effort at benchmarking AIS against RESuM, given our limited experience with AIS. More details about our effort in AIS design can be found in Appendix 14.
>
> **TL;DR:** 1. Our benchmark study across 10 LF simulation trials indicates that AIS performs worse than uniform sampling when averaged over all trials 2. We have faced significant challenges implementing AIS for our specific data type, while CNP provides a simple yet elegant solution 3. Additional AIS trials are currently running, but we are uncertain if we can produce a result by today.
>
> 1. **Our benchmark study across 10 LF simulation trials indicates that AIS performs worse than uniform sampling**: For this study, we selected 10 trials where both LF and HF simulation exist under the same sets of design parameters. We considered y_raw(HF) as our ground truth, and tested how well LF simulation trials (uniform sampling) and AIS trials (adaptive importance sampling) could reproduce our group truth. Please note that both the LF simulations and AIS simulations  was ran over exactly the same amount of events/neutrons (50,000), the only difference is the sampling strategy. The result is shown in the following Table:
>
> | Trial | y_raw(LF) | y_raw(AIS) | y_raw(HF ground truth) | MSE_LF | MSE_AIS |
> |-------|-----------|---------------|----------------------|---------|----------|
> | 0 | 0.32124 | 0.32051 | 0.23795 | 0.00694 | 0.00682 |
> | 1 | 0.13002 | 0.21761 | 0.15294 | 0.00053 | 0.00418 |
> | 2 | 0.14980 | 0.24280 | 0.12578 | 0.00058 | 0.01369 |
> | 3 | 0.23298 | 0.25374 | 0.24089 | 0.00006 | 0.00017 |
> | 4 | 0.13920 | 0.22771 | 0.14470 | 0.00003 | 0.00689 |
> | 5 | 0.15960 | 0.21050 | 0.16710 | 0.00006 | 0.00188 |
> | 6 | 0.18560 | 0.19835 | 0.19390 | 0.00007 | 0.00002 |
> | 7 | 0.15160 | 0.21796 | 0.18970 | 0.00145 | 0.00080 |
> | 8 | 0.16800 | 0.17575 | 0.17780 | 0.00010 | 0.00000 |
> | 9 | 0.14080 | 0.15976 | 0.15690 | 0.00026 | 0.00001 |
> | **Average over Trial 0-9** | | | | **0.00101** | **0.00345** |
>
> Although AIS surpasses LF simulation on a few trials(Trial 6,8,9, for example), when calculating the MSE over all 10 trials, we observed that LF simulation has smaller MSE than AIS simulations. We think this behavior is expected: as we pointed out in the previous reply, IS can struggle with discontinuous settings or irregular problem boundaries as they rely on smoothness in G(X). However, our problem turns out to be a highly stochastic and non-smooth setup.
>
>  2. **We have faced significant challenges implementing AIS for our specific data type, while CNP provides a simple yet elegant solution (More in Appendix 14):**  The AIS framework was designed to approximate a target distribution through iterative refinement of a proposal distribution. We leveraged Gaussian mixture models, kernel density estimation (KDE), and clustering techniques to dynamically adapt to the target distribution.
>
> * Initial Proposal Distribution: A Gaussian mixture model was used as the initial proposal distribution. Each component of the mixture is defined by a mean vector, a covariance matrix, and a weight:
>    * To obtain these, we tested between a K-means clustering algorithm and a KDE algorithm, and choose the better one (KDE) according to entropy and effective sampling size.
>    * When selecting the initial proposal distribution, we encountered rare event constraints: some LF trials contained only 5 triggered signals, limiting us to constructing the initial distribution from this small sample of signal events.
>
> * Sampling and Importance Weights:In each iteration, samples are drawn from the current proposal distribution. Importance weights are computed to adjust for the difference between the target distribution and the proposal distribution.
>    * We needed to incorporate pre-defined physics constraints into the AIS sampling process to ensure all samples remained within physically reasonable regions in LEGEND detector.
>    * To ensure numerical stability, we have to incorporate additional techniques such as dynamic mean shifts based on sample weights, PCA for dimensionality reduction of the covariance matrices, and rigorous covariance regularization.
>    * Additional fine-tuning is still needed for some trials.
> 3. **Additional AIS trials are currently running:** We are currently running the AIS algorithm over all 310 trials aligned with our LF simulation, so that we can benchmark it against RESuM model. This process takes time mainly because AIS works very well on some trials, but fails on some other trials even after fine-tuning. We hope to complete this study by today, and if we can finish before the response deadline, we will provide additional benchmarking results.

---

### Official Review · Reviewer_aEWx · 2024-11-04

**Soundness:** 3
**Presentation:** 2
**Contribution:** 3
**Rating:** 8
**Confidence:** 2

**Summary:**

This paper addresses a key question in physics: the matter-antimatter asymmetry in the universe, focusing on neutrinoless double-beta decay (NLDBD). The authors tackle the challenge of optimizing detector designs to minimize background event contamination, framing it as a Rare Event Design (RED) problem.
They introduce the Rare Event Surrogate Model (RESuM), which combines a pre-trained Conditional Neural Process with a Multi-Fidelity Gaussian Process to optimize detector designs. Applied to neutron moderator designs for the LEGEND NLDBD experiment, RESuM achieves a 66.5±3.5% reduction in neutron background while using only 3.3% of the computational resources of traditional methods. This innovative approach has broad implications for similar challenges in physical sciences.

**Strengths:**

The authors did a great job narrating each step of the procedure and model description. However, since I am not in the field, I got a little lost in the setup of the Experiment section, which involves applications. The results are convincing, though.

**Weaknesses:**

- It would be better if the authors provided more detailed explanations in the Experiment section.
- Several mathematical symbols are not in math mode.

**Questions:**

Are there no other works attempting to solve the same problem that we can use to benchmark your work against?

---

> ### Author Response · Authors · 2024-11-20
>
> We thank the reviewer's thorough evaluation and recommendation for acceptance. We have addressed the reviewer's feedback with careful revisions to the manuscript:
>
> > It would be better if the authors provided more detailed explanations in the Experiment section.
>
> Response: Thank you for your comment. To improve readability while preserving essential information, we revised Sections 5 and 5.1 by implementing the following changes. We hope these revisions have made the experimental section clearer and more accessible:
> * We rewrote the description of the simulation process in a more reader-friendly manner and included additional details
> * At the beginning of Section 5, we provided a clear, step-by-step outline of the three-stage physics process detailing how neutrons and $^{77}$Ge backgrounds are generated.
> * We aligned the simulation description with the parameters defined in Sections 3 and 4 (such as θ, ϕ, g(ϕ), m, and N) to “map” the statistical notations to the actual physics meaning.
>
> > Several mathematical symbols are not in math mode.
>
> Response: Thank you for pointing this out, we have addressed these in our new manuscript
>
> > Are there no other works attempting to solve the same problem that we can use to benchmark your work against?
>
> Response: Thank you for your comment. This work is based on the RED problem derived from LEGEND’s specific physics need, so we believe that we are the first to attempt this problem. Following the reviewer's suggestions, we have conducted a new benchmarking study, and we kindly redirect the reviewer to the General Updates of the Revised Manuscript on top of the page for more details.

---

> > ### Comment · Reviewer_aEWx · 2024-11-24
> >
> > Thank you for addressing my concern. I have now increased the score to 8.

---

> > > ### Author Response · Authors · 2024-11-24
> > >
> > > Thank you for the prompt reply and increase the score, we highly appreciate that. On behalf of all authors, we think that your comment has has significantly improved the quality of our manuscript.

---

### Official Review · Reviewer_Y2Co · 2024-11-04

**Soundness:** 4
**Presentation:** 4
**Contribution:** 4
**Rating:** 10
**Confidence:** 5

**Summary:**

The article presents a study on the optimal design of a neutrinoless double-beta decay detector, a Rare Event Design. The authors introduce a surrogate model, RESuM, to solve this optimization problem and gain computational cost. This algorithm is based on Conditional Neural Process, incorporating prior information. The authors applied their work to the LEGEND experiment, a physics detector for neutrinoless double-beta decay.

The Rare event Design problem is mainly well stated and very clear.  The Large and Small N scenarios provide a very good understanding of the problem.

Bayesian prior knowledge with Conditional Neural Process (CNP) is clear in the text. CNP is explained in Appendix 11. The details shows the CNP is rich : by the use of surrogate modelling it generates data as fast simulator. It is more efficient for exploration of parameter space, uncertainty quantification and model validation.

The simulations, results and validation of RESuM are well explained.

RESuM reduces LEGEND neutron background by (66.5 \pm 3.5)% using only 3.3% of the computational power of traditional methods. This is a very good achievement.

All in all, this is a very good paper.

**Strengths:**

The study is well conducted, very clear and concise, very interesting. The stated results are also very impressive and I hope this will lead to a larger breakthrough regarding neutrinoless double-beta decay community. The code is well presented and commented.

The work can be applied to Astronomy or Material Science. There are many other application possibilities in physics.

The work is motivated by the experimental discovery of neutrinoless double-beta decay (NLDBD), leading to answering the question ‘Why is there more matter than antimatter in our universe?’. Which makes a link with one of the opening questions of Javier Duarte talks at ICML24 ‘What is our universe made of?'. Thus, this work is very relevant for the community, and source of enthusiasm since the potential discovery of NLDBD would  lead to a Nobel-Prize-level breakthrough in physics.

**Weaknesses:**

The authors indicates in the limitations and applications part that due to limitation of ressources available they computed only 100 High Fidelity simulation, maybe the publication of the article will help to convince to gain more ressources.

The authors also add that the active learning functions used un RESuM are simplistic, they will consider more sophisticated one in future work.

Minor comments that do not impact the score :

They are many inconsistencies in the notations, here are what I noted while reading, some might be redundant

Line 95, independent of the other events

Line 97, \mathbf{Phi} is not defined. Is it for conciseness?

Line 97,  dollar i dollar-th

Line 133, dollar { y \in  blablabla } dollar should make it appear on the same line

Line 153-154, Why highlight +1 and +0? Should be in dollar 1 dollar, same for dollar 0 dollar

Line 186, \mathbf{\phi}

Line 187, the same

Line 188,  \mathbf for theta

Line 172, The use of the nuisance parameters is unclear to me. The vertical bar followed by X_ki, phi_ki, theta_k means conditionally, if possible this should be improved.

Line 183, Figure 1 title is ‘Overview of the RESuM framework’ which is not the case of Figure 3. I assume the authors mean Figure 1 then.

In Figure 1, if the MFGP combines prediction from LF and HF simulations, why are the arrows pointing at them and not starting from them?

In the section 3, the vector of design parameters is \mathbf{\theta} thus it should be the same in all the study.

In the introduction, line 49, 51, 52. According to section 4 it should be in \mathbf?

In the section 3,

Line 92-93, is it in \mathbf?

Line 96, in \mathbf

Line 108-109, the same

Line 113-114, the same for \theta and \phi

Line 120,  the same for \theta, N should be in  dollar N dollar

Line 124-125, the same for \theta

Line 126, the same for \theta twice in the eq

Line 126-127, \infty should be used and same for \theta

In the section 4,

Line 140, mathbf for \theta

Line 151, the same for \theta_k the k-th simulation trial

Line 152, the same for \theta_k and \phi_ki

Line 153, it should be  dollar N dollar,  dollar y dollar also

Line 160-161, \mathbf for \theta_k and \phi_ki

Line 187-188, \mathbf for \phi and \theta

Line 204, the same for \theta

Line 209, the same for \theta_k and \phi_ki

Line 212-213, the same for \theta and \phi_ki

In the section 5,

Line 261-263, the parameters of design should be the same as in the figure 2 (\varphi in particular). Figure 2 represents explicitly 4 parameters, n is missing

Line 265, \mathbf for the design parameters and the corresponding space

Line 289-290, \mathbf for \phi

Line 298, the same for \phi

Line 360, I suggest ‘plotted against the five parameters in Fig3.’ Since you only present five.

Line 370-371, the same for \theta

Line 395-400, parameters should be consistent with line 261-263

Line 404-405, parameters should be consistent with line 261-263

Line 409, also

Figure 4 and Table 1, should be consistent with line 261-263

Line 441-446, the same for \theta and the corresponding space

In section 6,

Line 491 and 492, the same for \theta and \phi

In section 2,

Line 93-93, I would have introduced the stochastic process X_1, …, X_N and specify after that

Line 213, \mathbf for theta and phi

Line 222, ^{HF} is necessary for y_CNP

Line 224, ^{LF} is necessary for y_CNP

In the supplementary materials

Line 738, \mathcal{L}_k appears two times

A.9 must be improved:

Are \theta and \theta’ vectors as in the manuscript?

What kind of matrix is K? What are K_LL, K_\delta?

Could you give a reference or improve the theory, please?

A.10 \Theta must be in bold

Is \theta a scalar or a vector?

Could you give a ref or a proof for the approximation of I(\theta) for GP model? It seems simple, but I would like to be convinced this is right.

**Questions:**

Line 53, this is interesting, where does this order of magnitude 'N needs to be extremely large (O(10^4))'come from? Could you give a quick explanation or a reference, please?

In part 5.0.2. why are relationship dependencies clear for R and N and not for the thickness and the angle? What does this implies on the model's interpretation or performance?

Line 171, the nuisance parameters are introduced quickly. Can you provide a brief explanation and give a reference, please?

---

> ### Author Response · Authors · 2024-11-21
>
> We would like to thank the reviewer for the thorough review and strongly support our paper to be accepted. We are deeply grateful for the reviewer's positive review of our work, which highlights the clarity, relevance, and potential impact of our manuscript. We have carefully revised our manuscript based on your comment:
>
> > Minor comments that do not impact the score :
>
> Response: we would like to thank the reviewer again for thoroghly reviewing our paper and giving many suggestions. We have carefully addressed all of them in our updated manuscript, below are some changes that may require additional explanation/clarification:
>
> * _Line 95, independent of the other events_: Added: Each random variable $X_i$ is statistically independent of all other $X_j$ where $j \neq i$
> * _Line 97, \mathbf{Phi} is not defined. Is it for conciseness? _ : We added a definition of $\mathbf{\Phi}$.
> * _Line 172, The use of the nuisance parameters is unclear to me. The vertical bar followed by X_ki, phi_ki, theta_k means conditionally; if possible this should be improved_:  $X_{ki}$ is the random outcome, while $\theta_{k}$ and $\phi_{ki}$ are the model parameters and event-specific parameters of the i$^{th}$ event in k$^{th}$ simulation trial. We added additional explanation about nuisance parameters.
> * _In Figure 1, if the MFGP combines prediction from LF and HF simulations, why are the arrows pointing at them and not starting from them?_: We updated Figure 1 to clarify these confusions, and we created both training and inference arrows to indicate the two different phase of model.
> * _Line 261-263, the parameters of design should be the same as in the figure 2 (\varphi in particular). Figure 2 represents explicitly 4 parameters, n is missing_: n means the number of panels in the panel design, we have modified Figure 2 to also include n.
> * _A.9 must be improved:_: we have improved Appendix 9 following the reviewer's suggestions.
> * _Could you give a ref or a proof for the approximation of I(\theta) for GP model? It seems simple, but I would like to be convinced this is right_: we addied a citation for this suggestion

---

> > ### Author Response · Authors · 2024-11-21
> > **Response to Reviewer's questions**
> >
> > > Line 53, this is interesting, where does this order of magnitude 'N needs to be extremely large (O(10^4))'come from? Could you give a quick explanation or a reference, please?
> >
> > Response: Thank you for pointing this out. The order of magnitude N=O(10^4)  is derived from our data, where the background neutron of interest occurs approximately once in 10^4 total background neutrons. This frequency guides the number of events N’>N required per single simulation run to ensure that the statistical noise in the simulation outcome is reduced to an acceptable level. However, it is important to note that this value is not a strict threshold for the rare event assumption but rather a practical choice informed by the rarity of the event in our data and the desired level of precision in our analysis. We also understand that presenting N=O(10^4) without context in the introduction may cause confusion, as it is not a universal value. To improve clarity, we removed this number from the introduction and replaced it with a more general statement "N needs to be very large for m to even be non-zero". We mentioned this number specifically in the experimental section (5.1 & 5.2), where it is more appropriate and contextually relevant.
> >
> > > In part 5.0.2. why are relationship dependencies clear for R and N and not for the thickness and the angle? What does this implies on the model's interpretation or performance?
> >
> > We thank the reviewer for pointing this out, as it touches on a critical aspect of RESuM model. The interpretability of r and n naturally arises from their trend with respect to the neutron moderation efficiency, reflecting well-understood physics:
> >
> > * r as the distance of the moderator from the detector array and it is related to the total amount of volume the moderator could cover
> > * n is the number of panel that forms the neutron moderator, and the gap between panels could allow neutron to slip through.
> >
> > The trend of the L, varphi, and d parameters is less obvious compared to the other parameters due to several factors: the stochastic nature of the system, the complex physics involved, and the limited understanding of how these parameters interact with each other and influence the overall design. We would like to point out that these design parameters have a more complex intercorrelation in high dimensional space. Our new benchmarking effort in Section 6 indicates that the RESuM model captures this correlation well on out-of-sample dataset.
> >
> > > Line 171, the nuisance parameters are introduced quickly. Can you provide a brief explanation and give a reference, please?
> >
> > Thank you for pointing this out. We have revised the text to provide a clearer explanation of the nuisance parameters. Specifically, we now clarify that the nuisance parameters, denoted as w , are the trainable parameters of the Conditional Neural Process (CNP), such as the weights and biases of the neural network. These parameters are optimized during training by minimizing the likelihood of the observed data. To address your request for a reference, we have cited the original CNP paper by Garnelo et al. (2018), which describes the framework and training methodology in detail.

---

> > > ### Comment · Reviewer_Y2Co · 2024-11-26
> > > **About Authors revisions regarding my questions**
> > >
> > > Regarding the order of magnitude of neutron events, it is now more explicit where the 10^4 comes from and implies rare events in the data of the authors. The general statement in the introduction makes sense.
> > >
> > > The explanation regarding the dependency on the thickness, the length and the angle is clear in the comments of the authors. If possible, could you briefly say some words on at the end of 5.2 or refer to another section?
> > > The section 6 is also very relevant to the study.
> > >
> > > Yes, the reference of Garnelo et al. 2018 for the CNP parameters is better for the reader.
> > >
> > > Here is also a minor revision I found :
> > >
> > > -Line 465, I assume this is \hat{y}_{Raw}
> > >
> > > -Line 465-466, maybe using dollar { 68.27 backslash% } dollar would help to put everything on the same line, I hope there won't be line continuation..

---

> > ### Comment · Reviewer_Y2Co · 2024-11-25
> > **Answer on minor revisions**
> >
> > Thank you ofr taking into account my comments. This points are more clear now.
> >
> > Here are some minor things I have seen quickly :
> >
> > Line 103, 'i-th' might be dollar i dollar-th. There are still some inconsistencies regarding the printing of these as seen at line 164 and 188. This is purely aesthetic.
> >
> > A point is missing at line 241.
> >
> > In the appendix 10, a \mathbf{\Theta} is missing at line 725

---

> > > ### Author Response · Authors · 2024-11-25
> > >
> > > Thank you for the prompt reply and additional comments. We have corrected all these, and we will upload a new version of the manuscript before the review deadline. On behalf of all authors, we would like to sincerely thank you again for strongly recommending this paper to be accepted.

---

> ### Author Response · Authors · 2024-11-26
>
> Thank you for the additional comments, we have addressed all of them in our revised manuscript. We plan upload the new version tomorrow (Nov. 27th) :
>
> > The explanation regarding the dependency on the thickness, the length and the angle is clear in the comments of the authors. If possible, could you briefly say some words on at the end of 5.2 or refer to another section? The section 6 is also very relevant to the study.
>
> Response: we added this into the updated draft when discussing the trend plot:
>
> _The trends shown in Figure 4 reflect well-known detector physics: $r$ represents the moderator-detector distance, which determines the moderator's coverage volume, while $n$ denotes the number of moderator panels, where inter-panel gaps allow neutrons to escape. The dependence on other parameters are less obvious due to the schocastic nature of the system._
>
> > Line 465-466, maybe using dollar { 68.27 backslash% } dollar would help to put everything on the same line, I hope there won't be line continuation..
>
> Response: thank you for pointing this out, we changed our LaTeX code this way and now they are in the same line.

---

### Official Review · Reviewer_BhVu · 2024-11-05

**Soundness:** 3
**Presentation:** 4
**Contribution:** 3
**Rating:** 8
**Confidence:** 3

**Summary:**

The paper presents a surrogate model for rare events in physics.  The model is a hybrid model combining a conditional neural process (CNP) model and a multi-fidelity Gaussian process (MFGP).  The CNP is pre-trained on both low-fidelity (LF) and high-fidelity (HF) simulation data.  Subsequently the CNP model is used to calculate additional design data (averaged CNP scores for LF and HF data).  The MFGP model is initially trained using HF, HF CNP and LF CNP data, and further tuned using Bayesian Optimization (BO) equipped with the integrated variance reduction acquisition function to minimize the total variance of the model.

**Strengths:**

The RESuM model is well-motivated and the application in physics is quite interesting.  While I am not a statistician nor familiar with the CNP model, I am very familiar with GP models and BO design, and can see no obvious mistakes in the paper.  Moreover the results are thorough and I can see a clear application for this model outside of that considered here.

**Weaknesses:**

It appears that the real goal of this paper is to optimise particle detector design.  As such, it strikes me that the surrogate model need only be accurate in those regions of design space that are "good" - ie minimise background noise.  However the RESuM model is designed to model the entire design space, including "bad" designs (eg it may request HF simulations for regions whose prior confidence bounds are such that we know whp will not be appropriate in practice).

As such perhaps a more efficient approach would be to replace the integrated variance reduction acquisition function used in the BO portion of the design with a more goal-oriented acquisition function like expected improvement (EI) or GP-UCB, which would naturally focus computational effort on optimizing the detector design (ie the real underlying goal) rather than modeling all possible detectors (essentially the first step in the current design procedure).

(Note I intend this more as a point for discussion rather than a major criticism - the current approach is valid, but perhaps the strong focus on modeling is obscuring the real goal).

**Questions:**

In addition to the above:

- I am a little unclear regarding the role of CNP here.  Is it primarily a means to include the LF simulations in the overall model in a way that takes advantage of the physical insight in the combined LF/HF data?
- in the active learning / Bayesian optimization phase you only run HF simulations.  Have you considered using multi-fidelity BO [1,2] to take advantage of the cheaper LF simulations as well?
- As a minor note, on line 53 you say that N is $\mathcal{O} (10^4)$.  Do you mean $N$ is of order $10^4$?

---

> ### Author Response · Authors · 2024-11-21
>
> We thank the reviewer for their thorough review and support for paper acceptance, particularly for highlighting that 'the RESuM model is designed to model the entire design space, including "bad" designs.' Indeed, the current RESuM model aims to learn a comprehensive mapping f: θ→y across all possible designs, regardless of good/bad. This learned mapping then enables us to identify optimal designs that minimize the design metric y. The uncertainty-aware properties of MFGP further enhance the robustness of our design optimization approach. We have clarified this mapping objective in our updated manuscript.
>
> As part of the future work, we are currently exploring advanced acquisition functions in the active learning phase to enhance the efficiency of discovering promising designs. The Expected Improvement (EI) the reviewer brought up is exactly one of our primary focus. We think the RESuM's current approach of learning across the complete design space offers valuable interpretability benefits. As demonstrated in Figure 4, by modeling both successful and unsuccessful designs, RESuM reveals important physical trends—for instance, how the performance metric decreases with larger radius up to 160cm, beyond which further increases yield no further gain. Such insights help traditional, non-ML physicists  to understand the model's decision-making process.
>
> > I am a little unclear regarding the role of CNP here. Is it primarily a means to include the LF simulations in the overall model in a way that takes advantage of the physical insight in the combined LF/HF data?
>
> Response: The primary goal of CNP is to smooth out the highly discrete output of each simulated event. Without CNP, each simulated event can only yield 1 (a signal is created) or 0 (no signal created). CNP allows us to map each event into a floating point number between 0 and 1 (i.e. 0.8 could mean this event is 80% likely to create a signal), thereby smoothing out the discreteness in the output. Also as the reviewer pointed out, CNP is independent of fidelities, thereby, it also serves as a link to combine HF/LF data. Physical insight
>
> > in the active learning / Bayesian optimization phase you only run HF simulations. Have you considered using multi-fidelity BO [1,2] to take advantage of the cheaper LF simulations as well?
>
> Response: This is an excellent suggestion that aligns with ongoing discussions among our team. We are exploring the potential of MFBO to guide both the selection of additional low-fidelity simulations or high-fidelity simulations with EI. This aligns with our ongoing work in improving the active learning part of this algorithm. Such an approach could significantly enhance our method's efficiency.
> * As a side note, we believe the reviewer meant to provide two citations [1,2] but the papers were not shown in the review. We would love to learn additional works on MF-BO direction to further improve our model.
>
> > As a minor note, on line 53 you say that N is O(10^4). Do you mean N is of order 10^4?
>
> Response: Thank you for pointing that out, yes this is what we mean. Following other reviewer's comment, we think providing this number in the introduction only adds additional confusion, so we decided to remove it and rephrase our sentence in a more general form: "Due to the ultra-pure nature of the NLDBD detector, N needs to be very large for m to even be non-zero"

---

> > ### Comment · Reviewer_BhVu · 2024-11-25
> > **Response**
> >
> > Thank you for your clarifications, I am happy to keep my recommendation as is.  The references I forgot to include were Kandasamy's works on Multi-fidelity Bayesian optimization (see eg ICML2017), but there are others in the similar vein from around this time.

---

> > > ### Author Response · Authors · 2024-11-25
> > >
> > > Thank you for your prompt response and keeping your recommendation. We will check out those paper and using those to improve our model.

---

### Author Response · Authors · 2024-11-20
**General Updates of the Revised Manuscript**

We would like to thank all reviewers for spending time reading our paper and providing their valuable feedbacks. We have prepared a new version of the manuscript and a general response to all reviewers. We respectfully invite reviewers to check out the new version. Here are some general updates we have made to the manuscript:
* **Model Benchmarking**: This work is based on the RED problem derived from LEGEND’s specific physics need, therefore we believe that we are the first to attempt this problem. In the original manuscript, we conducted a basic benchmarking comparison between RESuM (MFGP+CNP) and an MFGP model (see Section 5.0.4 and Appendix 12 of old manuscript). Following reviewer feedback, we have conducted a more comprehensive model evaluation, with detailed technical aspects of model selection, training data preparation, and performance metrics now documented in Appendix 13: Model Benchmarking Details. The complete benchmarking results are presented in Section 6 (Model Validation and Benchmarking) and Table 2:
| Trial | Model | Dataset (#LF,#HF) | MSE | 1σ̂ [%] | 2σ̂ [%] | 3σ̂ [%] | Cost [%] |
|-------|-------|------------------|-----|---------|---------|---------|-----------|
| 1 | MF-BNN | (305,5) | 0.471 | 100 | 100 | 100 | 1.7 |
| 2 | MF-BNN | (307,7) | 0.439 | 100 | 100 | 100 | 2.3 |
| 3 | MF-BNN | (310,10) | 0.021 | 100 | 100 | 100 | 3.3 |
| 4 | MFGP | (305,5) | 0.025 | 42 | 100 | 100 | 1.7 |
| 5 | MFGP | (307,7) | 0.007 | 31 | 56 | 79 | 2.3 |
| 6 | MFGP | (310,10) | 0.015 | 17 | 33 | 48 | 3.3 |
| 7 | RESuM | (305,5) | 0.002 | 67 | 96 | 99 | 1.7 |
| 8 | RESuM | (307,7) | 0.003 | 64 | 92 | 99 | 2.3 |
| 9 | RESuM | (310,10) | 0.002 | 69 | 95 | 100 | 3.3 |
| 10 | RESuM (100x) | (310,10) | 0.003 | 62.4 | 92.2 | 99.6 | 3.3 |
| | Proper Statistical Coverage | | | 68.27 | 95.45 | 99.73 | |
* Section 2: Related Works has been expanded to include a comprehensive discussion of rare event simulation and modeling, along with 11 relevant citations. We also discussed the difference between our proposed RED problem and conventional rare event simulation/modeling problem in reliability engineering.
* The first part of Section 5: EXPERIMENT AND RESULT is rewritten to facilitate non-physicists readers.
* Previous Subsection 5.0.4 SURROGATE MODEL VALIDATION is promoted to new Section 6: MODEL VALIDATION AND BENCHMARKING
* Appendix 12: COMPARATIVE ANALYSIS OF MULTI-FIDELITY APPROACHES is removed, its content is absorbed into new Section 6 and Appendix 13.
* We edited the statistical notation and typos following reviewer’s comment.

We are working on individual responses to each reviewers, and these will be posted in the next 1-2 days.

---

> ### Author Response · Authors · 2024-11-28
> **Follow-Up General Updates**
>
> Dear all,
>
> Thank you for the comment and feedback. We have uoloaded a new version of the manuscript incorporating the following items:
>
> * Addressed additional comments (mostly editorial) from reviewers in the second/third round
> * Added Appendix 14: Adaptive Importance Sampling section to discuss technical details and challenges in our Adaptive Importance Sampling (AIS) benchmarking effort.
>
> We spent a significant amount of time in AIS benchmarking and encountered several challenge in developing the AIS algorithm. The challenges were discussed in details in the newly added Appendix 14. We still hope to complete the study by December 2nd and include our benchmarking results in the previous official comments.
>
> Sincerely,
>
> The Authors

---

### Comment · Area_Chair_Avug · 2024-11-26

Dear all,

The deadline for the authors-reviewers phase is approaching (December 2).

@For reviewers, please read, acknowledge and possibly further discuss the authors' responses to your comments. While decisions do not need to be made at this stage, please make sure to reevaluate your score in light of the authors' responses and of the discussion.

- You can increase your score if you feel that the authors have addressed your concerns and the paper is now stronger.
- You can decrease your score if you have new concerns that have not been addressed by the authors.
- You can keep your score if you feel that the authors have not addressed your concerns or that remaining concerns are critical.

Importantly, you are not expected to update your score. Nevertheless, to reach fair and informed decisions, you should make sure that your score reflects the quality of the paper as you see it now. Your review (either positive or negative) should be based on factual arguments rather than opinions. In particular, if the authors have successfully answered most of your initial concerns, your score should reflect this, as it otherwise means that your initial score was not entirely grounded by the arguments you provided in your review. Ponder whether the paper makes valuable scientific contributions from which the ICLR community could benefit, over subjective preferences or unreasonable expectations.

@For authors, please respond to remaining concerns and questions raised by the reviewers. Make sure to provide short and clear answers. If needed, you can also update the PDF of the paper to reflect changes in the text. Please note however that reviewers are not expected to re-review the paper, so your response should ideally be self-contained.

The AC.

---

### Author Response · Authors · 2024-12-03
**Summary of Review and Responses**

Dear Reviewers and Area Chair,

Thank you for the precious time you put into the review of this paper. On behalf of all authors, we think your comments has helped making this paper much better. Below is a summary of rebuttal process:

The majority of the reviewers have recommended acceptance of this paper. The reviewers found the work to be well-motivated (BhVu, Y2Co, TJgd), well-conducted with no obvious mistakes. (Y2Co, aEWx, BhVu, TJgd). Reviewers pointed out that the integration of CNP and MFGP is a novel approach for handling rare event problems (TJgd). Reviewers also commented that this work is interesting to the phsics community (BhVu, Y2Co) and other scientific domains (Y2Co, TJgd).. The reviewers particularly praised the clear, systematic presentation of the methodology and model description (Y2Co, aEWx). All reviewers provided detailed editorial comments and clarification questions, which have been thoroughly addressed in the revised manuscript. Notably, Reviewers aEWx and KsYG pointed out the need for baseline comparisons, leading to additional benchmarking studies in the updated manuscript. Both reviewers increased their score on the updated manuscript. In response to KsYG's request for benchmarking against Adaptive Importance Sampling~(AIS) algorithms, we implemented this comparison and presented the results in our [latest response](https://openreview.net/forum?id=lqTILjL6lP&noteId=rjtjnVChWc), which shows that RESuM is still advantageous on our specific data type.

Thanks again,
Authors

---

### Meta-Review · Area_Chair_Avug · 2024-12-19

**Metareview:**

The reviewers recommend acceptance (8-10-8-5-6). The paper presents a well-motivated approach for the design of particle physics detectors. The presentation is clear and, according to at least one reviewer, the results are impressive. The author-reviewer discussion has been constructive and has led to a number of improvements to the paper, in particular regarding its presentation and the discussion of related work. Reviewer KsYG recommends further comparison with baselines, in particular with Importance sampling-based optimization methods. Results discussed during the author-reviewer discussion period seem to indicate that Adaptive importance sampling does not perform as well as a uniform baseline. For these reasons, and given the otherwise positive reviews, I recommend acceptance. I encourage the authors to address the remaining concerns in the final version of the paper.

**Additional Comments On Reviewer Discussion:**

The discussion has been constructive and has led to several improvements to the paper.

---

### Decision · Program_Chairs · 2025-01-22

Accept (Spotlight)